# GIVE: Structured Reasoning of Large Language Models with Knowledge-Graph-Inspired Veracity Extrapolation

**Jiashu He** [1]   **Mingyu Derek Ma** [2]   **Jinxuan Fan** [3]   **Dan Roth** [1]   **Wei Wang** [2]   **Alejandro Ribeiro** [1]

## Abstract

Existing approaches based on context prompting or reinforcement learning (RL) to improve the reasoning capacities of large language models (LLMs) depend on the LLMs' internal knowledge to produce reliable Chain-Of-Thought (CoT). However, no matter the size of LLMs, certain problems cannot be resolved in a single forward pass. Meanwhile, agent-based reasoning systems require access to a comprehensive nonparametric knowledge base, which is often costly or not feasible for use in scientific and niche domains. We present Graph Inspired Veracity Extrapolation (GIVE), a novel reasoning method that merges parametric and non-parametric memories to improve accurate reasoning with minimal external input. GIVE guides the LLM agent to select the most pertinent expert data (**observe**), engage in query-specific associative thinking (**reflect**), and then synthesize this information to produce the final output (**speak**). Extensive experiments demonstrated the following benefits of our framework: (1) GIVE increases the performance of LLMs across various sizes. (2) In some scenarios, GIVE allows smaller LLMs to surpass larger, more sophisticated ones in scientific tasks (**GPT3.5T + GIVE** > **GPT4**). (3) GIVE is effective on scientific and open-domain assessments. (4) GIVE is a training-free method that enables LLMs to tackle new problems that extend beyond their training data (up to **43.5%** → **88.2%** accuracy improvement). (5) GIVE allows LLM agents to reason using both restricted (very small) and noisy (very large) knowledge sources, accommodating knowledge graphs (KG) ranging from **135** to more than **840k** nodes. (6) The reasoning process involved in GIVE is fully interpretable. Our code is available at **https://github.com/Jason-Tree/GIVE**

## 1. Introduction

Context-based methods (Wei et al., 2023; Brown et al., 2020) to enhance the reasoning ability of large language models (LLMs) (Ouyang et al., 2022; Chowdhery et al., 2022; OpenAI et al., 2024b; Grattafiori et al., 2024; OpenAI et al., 2024a) incorporate examples with logic chains in the prompt, allowing the model to generate analogous logical sequences for the given query. Recent investigations (Zelikman et al., 2024; DeepSeek-AI et al., 2025; Shao et al., 2024) highlighted the substantial promise of reinforcement learning (RL) in improving the generation of high-quality logic. These methods operate on the premise that the parametric memory of LLM is sufficient to execute accurate reasoning. They demonstrate satisfactory performance in reasoning-complex tasks that do not require novel knowledge, such as mathematical and common sense. However, they fail to achieve comparable enhancements in scientific tasks due to the insufficiency of pre-trained knowledge (Cai et al., 2024; Zhang et al., 2024; Dorfner et al., 2024; Dong et al., 2023; Zhong et al., 2023). This type of information is rare in web data; in fact, regardless of the number of parameters, LLMs face unsolvable challenges using a single forward pass (Dziri et al., 2024). Incorporating external knowledge becomes necessary to enable the model to adapt to tasks beyond the training set.

Recent studies have highlighted the significant potential of incorporating structured knowledge bases (KGs) into LLM inference processes to enhance knowledge provision. Advanced retrieval-augmented generation (RAG) frameworks (Liang et al., 2024; Panda et al., 2024) identify precise KG from documents to aid accurate information retrieval for specific queries; research efforts are also proposed to promote iterative exploration (Sun et al., 2024; Luo et al., 2024; Xu et al., 2024) and improved modeling of interconnections within knowledge bases to improve coherence (Mavromatis & Karypis, 2024; Edge et al., 2024).

[1]Department of Computer and Information Science, University of Pennsylvania, Philadelphia, United States [2]Department of Computer Science, University Of California, Los Angeles, Los Angeles, United States [3]Department of Statistics, University of California, Berkeley, Berkeley, United States. Correspondence to: Jiashu He <jiashuhe@seas.upenn.edu>.

*Proceedings of the 42nd International Conference on Machine Learning*, Vancouver, Canada. PMLR 267, 2025. Copyright 2025 by the author(s).

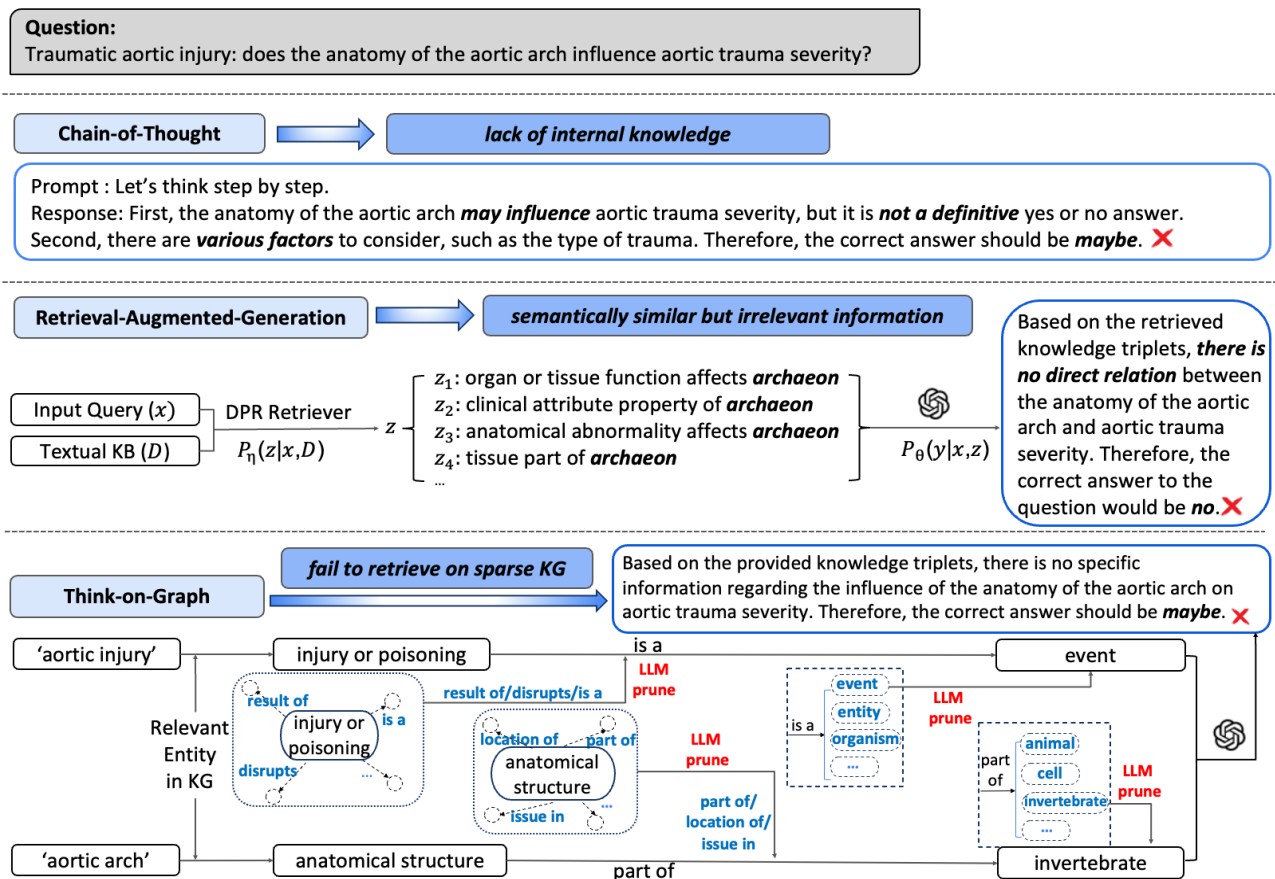

**Figure 1:** Without gold context, Chain-of-Thought (CoT) fails because LLM's internal knowledge fails to form a faithful logic chain; Retrieval-Augmented-Generation (RAG) retrieves semantically similar but irrelevant information, leading to hallucination; Think-on-Graph (ToG) focuses on using LLM to prune the direction of traverse, thus fails for lack of high-quality candidates.

These strategies have proven effective in specific question answering scenarios, assuming the existence of an accessible and retrievable database containing the correct reasoning logic. However, in scientific realms, the construction of comprehensive knowledge bases is daunting, which requires progress in both domain-specific natural language processing (NLP) and field-wide vocabulary standardization (Badal et al., 2019; Verhagen et al., 2012). Enhancing LLM reasoning with limited external information is a more realistic approach. Although the retrieved knowledge lacks direct evidence for problem solving, it embodies the intuition and expertise of curation specialists, such as feasible relation collection and potential links among similar entities.

In this paper, we seek to overcome the constraints of relying solely on either internal or external knowledge by introducing GIVE, a graph-inspired veracity extrapolation framework. GIVE emulates the cognitive processes employed by human experts, draws inspiration from related knowledge, and conducts associative thinking. It populates the limited information with provisional connections be-

tween query concepts, informed by factual linkages, and grounds these links using LLMs' parametric knowledge. Our method also constructs counterfactual reasoning to mitigate hallucinations and incorporates intermediate entities for multi-step reasoning. Specifically, GIVE identifies a concentrated group of entities closely associated with the query. By exploring potential relations within pertinent knowledge graph (KG) concepts, we develop a reasoning framework that encompasses all conceivable concepts and connections that could enhance query resolution. We incorporate additional intermediate concepts by selecting multi-step reasoning strategies that most effectively address the questions. GIVE comprises retrieved expert information, internal knowledge acquired through pre-training, and innovative relations that unite analogous concepts via veracity extrapolation. To complete the reasoning framework, counterfactual links among nodes are integrated to avert hallucinations. Ultimately, GIVE is a structured reasoning framework of LLMs that (1) retrieves external knowledge to increase the informativeness of responses; (2) conduct diver-

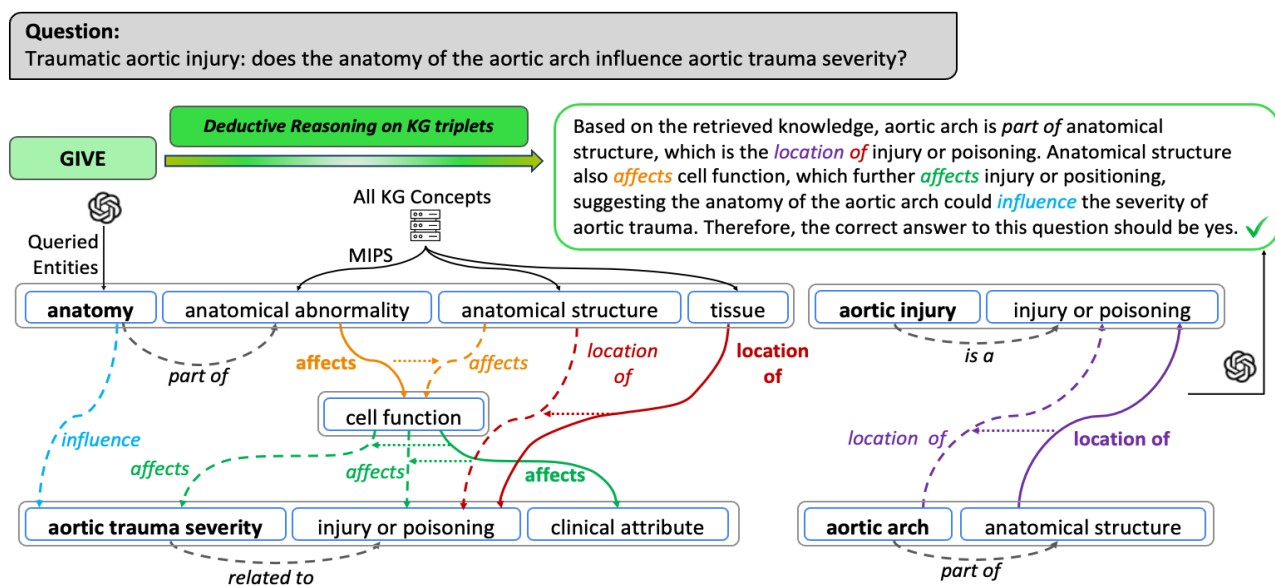

**Figure 2:** Reasoning process of GIVE. Solid lines indicate expert information, while dashed lines depict results from the "veracity extrapolation" procedure. GIVE first constructs an entity group for each queried concept, then induce inner-group connections using its internal knowledge. The expert's cross-group connections serve as evidence, guiding the LLM to extrapolate the veracity of potential relationships among similar concepts.

gent reasoning on the limited expert information that does **not** directly solve the query, by extrapolating expert triplets to correlated queried concepts, termed "veracity extrapolation." We test our proposed approach on both scientific and open-domain tasks, including biomedical, common-sense, and open-domain question answering. GIVE consistently achieves superior performance across all datasets, utilizing KGs of differing sizes and densities, surpassing all internal/external knowledge reasoning benchmarks, which demonstrates the effectiveness and reliability of the proposed framework. Encouragingly, GIVE enables smaller LLMs such as GPT3.5T to achieve better performance than advanced models like GPT4 in scientific tasks. GIVE pioneers in elevating LLM's reasoning capabilities with minimal external cues to activate its intrinsic problem-solving capacity.

## 2. Problem Statement

Assume that $\mathcal{R}(x)$ represents the accurate logic chain to solve the query $x$. Reasoning on internal knowledge techniques (Wei et al., 2023; Brown et al., 2020; DeepSeek-AI et al., 2025) make use of reinforcement learning or context-based prompts to direct the LLM to produce an accurate logic chain with high probability, relying solely on the model's intrinsic knowledge. This signifies a dependence on the parameter $\theta^*$, such that rationale$_{\theta^*}(x) \to \mathcal{R}(x)$. In contrast, reasoning with retrieval frameworks (Sun et al., 2024), posits the availability of a comprehensive knowl-

edge base $\mathcal{B}^*$, and employs a retrieval model $\beta$ such that retrieve$_\beta(\mathcal{B}^*, x) \to \mathcal{R}(x)$.

In this work, we generalize the benefits of both methodologies deliberately **not** presuming the existence of $\theta^*$ or the comprehensive nature of an accessible knowledge base $\mathcal{B}^*$. We utilize retrieved structured evidence to template problem solving, directing LLM to leverage its intrinsic knowledge on scientific tasks backed by minimal external information. GIVE is a training-free framework that directs LLM to conduct reasoning on the limited external information that does not directly solve the query:

$$\text{rationale}_{\text{GIVE}}(x, \text{retrieve}_{\text{GIVE}}(B, x)) \to \mathcal{R}(x) \quad (1)$$

While theoretically, this offers no benefit over RAG (Lewis et al., 2021) for generating answers with retrieved context, numerous studies (Ge et al., 2023; Bang et al., 2023) have shown that LLMs lack expertise in scientific domains and cannot construct a multi-step logical chain (Wei et al., 2023; Wang et al., 2023; Jiang et al., 2024) to connect a complex query with the retrieved incomplete information that cannot directly solve it. GIVE, in this context, provides additional guidance to LLMs, helping them to populate the expert knowledge towards the key components of the query by integrating both parametric and non-parametric knowledge.

# 3. Method

For structured information extraction, our study employs Knowledge Graph (KG) as the external knowledge base. A KG is defined as $G = \{E_G, R_G, \mathcal{E}_G\}$, where $\mathcal{E}_G = \{(u,r,v), u,v \in E_G, r \in R_G\}$, with $E_G$ being the set of entities and $R_G$ the set of relations. GIVE prompts inductive reasoning by: 1) breaking down the query into fundamental concepts and attributes; 2) forming entity groups by linking the queried entities with related concepts that exist in the KG; 3) inducing connections within groups between the queried entity and related concepts using LLM's parametric knowledge; 4) establishing affirmative and counter-factual inter-group links by probing and refining potential connections, and leveraging intermediate concept groups for multi-hop reasoning in complex queries. Appendix 1 provides a comprehensive explanation of the GIVE algorithm, while Appendix D includes examples of each subroutine.

## 3.1. Query Information Extraction

For a given query $x$, GIVE initially utilizes the LLM to extract the entity and relation sets $E_x$ and $R_x$, as represented by:

$$x \rightarrow LLM \rightarrow E_x, R_x \qquad (2)$$

Here, $E_x = \{e_x^0, e_x^1 ... e_x^n\}$ represents the top-k concepts, while $R_x = \{r_x^0, r_x^1 ... r_x^m\}$ comprises the top-m relations or attributes identified in the query. For example, the query "Is melatonian effective for insomnia?" contains entity set $E_x$ = {melatonin, insomnia} and relation set $R_x$ = {effective for}.

## 3.2. Entity Group Construction

This step aims to assist LLM to induce connections between queried entities from expert connections between similar KG entities. We do this by exploring the knowledge space to form a cluster of related concepts for each significant entity identified in the query. For each $e_x^k \in E_x$, GIVE uses a pre-trained LM encoder $w$ to encode knowledge base concepts and find $p$ most similar concepts to each entity in the latent space by comparing cosine similarities:

$$Y_x^k = \{y_{x1}^k, y_{x1}^k ... y_{xp}^k\} = \underset{\hat{y} \in E_G}{\operatorname{argmin}_p} \{cos(w(e_x^k), w(\hat{y}))\} \qquad (3)$$

The set $Y_x^k$ includes entities semantically akin to $e_x^k$, and $e_x^k$ is added to $Y_x^k$ to create the entity group $N_x^k$:

$$N_x^k = \{e_x^k\} \cup Y_x^k \qquad (4)$$

Relating queried concepts with the semantically similar KG entities broadens the reasoning scope from strict information retrieval to inferring relationships over a broader range of relevant concepts.

## 3.3. Inner-group connections

Focusing on each $N_x^i$ in Section 3.2, we establish links between the queried entity and its semantically related concepts within its entity group. This aims to prompt the LLM to explore related concepts broadly, rather than focusing solely on the queried entity. The challenge of directly inducing relationships between two queried entities is addressed by identifying possible links between two sets of similar concepts. We employ the LLM to suggest relationships between the queried entity and each in-group concept. All such knowledge are appended to an affirmative knowledge set. Consider an entity group with 1 queried entity and $p$ additional KG concepts $N_x^k = \{e_x^k, y_{x1}^k, y_{x2}^k, ..., y_{xp}^k\}$, for $1 \le m \le p$:

$$(e_x^k, y_{xm}^k) \rightarrow LLM \rightarrow (e_x^k, r_{xm}^k, y_{xm}^k) \qquad (5)$$

## 3.4. Inter-group connections

### 3.4.1. POTENTIAL RELATIONS INDUCTION

We first determine all possible relationships that could link any pair of nodes within the two groups. We evaluate two categories of potential relations: (1) **Relations specified in the question.** These are the vital connections required for the ultimate QA task. (2) **KG relations present between these groups.** Due to the semantic similarity of concepts within each group, existing cross-group KG connections may link other entity pairs across groups. Formally, for KG $G$, entity groups $N_i$ and $N_j$, we define the set of potential relations $R^{ij}$ as:

$$R_G^{ij} = \{r, (u,r,v) \in \mathcal{E}_G, u \in N_i, v \in N_j\} \quad R^{ij} = R_x \cup R_G^{ij} \qquad (6)$$

### 3.4.2. INTERMEDIATE NODE GROUP DISCOVERY FOR MULTI-STEP REASONING

In scientific fields, directly establishing links between specific concepts is sometimes not feasible. For instance, to address a query regarding a drug's effect on a disease, a logical approach is to associate these through (drug, compound, disease) links, forming the statement: "since a certain compound exists within the drug, and this compound alleviates particular diseases, it can be inferred that the drug may treat the disease." To support such multi-step reasoning process, GIVE investigates novel node groupings used as intermediate steps in reasoning. GIVE directs LLM to identify the most beneficial 2-hop paths between groups, helping with the question-answering task. Through the intermediate node of an optimal 2-hop path, GIVE forms an intermediate entity group using the process outlined in Section 3.2.

### 3.4.3. STRUCTURED EXPERT KNOWLEDGE GUIDED REASONING

Finally, GIVE introduces associative thinking from these non-parametric evidence, including concept groups and all potential relations between them.

**Relation Assignment Using External Evidence.** If an edge exists in the external knowledge graph between two entities, we inherit the ground truth relationship from the original KG. When verbalizing such edges, we use a definite tense to convey the high certainty of this fact.

**Veracity Extrapolation with Internal Knowledge.** Potential relations between node groups identified in Section 3.4.1 help to suggest possible node connections. It is vital to solidify the internal knowledge to validate the model's decision or discard incorrect answers with clear context. GIVE, therefore, instructs the LLM to assess each potential edge between node groups. Confirmed connections are added to the extrapolated affirmative knowledge set, while rejected ones offer counterfactual insights to avoid model hallucination.

**Identifying Open Relations for Novel Connections** To avoid the limitations of the knowledge base's scope, GIVE also offers LLM the freedom to independently generate a short phrase describing the relation of each entity pair.

### 3.5. Progressive answer generation

From the reasoning process presented in Section 3.4.3, GIVE obtained three types of knowledge: (1) Affirmative knowledge set that contains all knowledge affirmed by LLMs' parametric knowledge, including inner-group connections; the cross-group open connections, and the extrapolated connections. We refer to this knowledge set as $\tilde{\mathcal{R}}^{a}(x)$. (2) Counterfactual knowledge set that contains all rejected potential cross-group connections, which is referred to as $\tilde{\mathcal{R}}^{c}(x)$, (3) Ground-truth KG knowledge set $\tilde{\mathcal{R}}^{e}(x)$. To prevent hallucination, we adopt a progressive manner to generate the final answer by first giving only the affirmative knowledge set. GIVE further directs LLM to refine this answer with the full context and the counter-factual knowledge set. The final answer is generated by providing all previous context and the ground-truth knowledge set. Given generator $p_{\gamma}$, and the retrieved knowledge sets described above, the answer generation process by GIVE is defined as:

$$\text{GIVE}_{a}(y^{a}|x) := p_{\gamma}(y^{a}|x, \tilde{\mathcal{R}}^{a}(x)) \tag{7}$$

$$\text{GIVE}_{a+c}(y^{a+c}|x, y^{a}) := p_{\gamma}(y^{a+c}|x, \tilde{\mathcal{R}}^{a}(x), y^{a}, \tilde{\mathcal{R}}^{c}(x)) \tag{8}$$

$$\text{GIVE}_{a+c+e}(y^{a+c+e}|x, y^{a}, y^{a+c})$$
$$:= p_{\gamma}(y^{a+c+e}|x, \tilde{\mathcal{R}}^{a}(x), y^{a}, \tilde{\mathcal{R}}^{c}(x), y^{a+c}, \tilde{\mathcal{R}}^{e}(x)) \tag{9}$$

where $\text{GIVE}_{a}$, $\text{GIVE}_{a+c}$, $\text{GIVE}_{a+c+e}$ corresponds to the answer generation process that uses (1) only affirmative knowledge; (2) both affirmative knowledge and counterfactual knowledge; (3) the whole set of knowledge, which contains affirmative, counterfactual, and ground truth KG knowledge.

### 3.6. Running time

Consider $m$ entity groups with $n$ concepts each and $r$ candidate relations between two groups. Inner-group connections (Section 3.3) require $\mathcal{O}(mn)$ LLM calls. For inter-group connections (Section 3.4.3), the LLM calls needed are equal to the candidate potential connections for "veracity extrapolation", $\mathcal{O}(rm^{2}n^{2})$. As shown in Section 4.7, GIVE performs best with $n = 1$ or 2. Appendix C.1 further shows: (1) $m$ averages 3 or 4 for all datasets; (2) $r$ is bounded by 4 across datasets; (3) GIVE's running time and context length stay reasonable as KG size or sparsity increases; (4) GIVE's complexity can be further decreased by batch pruning and adding summarization agents to shorten knowledge.

## 4. Experiments

### 4.1. Research Questions

The experiments in this section aim to address the following research questions: (1) Is GIVE **robust** in enhancing LLMs' reasoning with a limited (very small) or noisy (very large) external knowledge base? We explore this in Section 4.4 and Section 4.5 using knowledge graphs ranging from 135 to over 840k entities, with statistics in Table 2. (2) Can GIVE improve reasoning across both **general and specific domains**? We test biomedical reasoning in Section 4.4, commonsense reasoning in Section 4.5, and open-domain reasoning in Section 4.6. (3) What are the impacts of GIVE's **different components, parameters** and implementation choices? This is examined through ablation studies in Sections 4.7.1, 4.7.2 and in Appendices B.2, B.1, and B.3. (4) What other factors might affect the performance of GIVE? We provide a comprehensive analysis in Section 4.7.3, discussing what makes effective "inspirations". (5) How does GIVE's **cost** compare to other methods? A detailed examination of run time and context length is in Appendix C.1.

### 4.2. Experiment Settings

We examine questions that **require additional reasoning** by disregarding any "gold" context or inclusive knowledge base. In PubmedQA (Jin et al., 2019), we challenge competing methods by supplying LLM only with the question statement and the retrieved facts, **excluding** any ground-truth context where the answer is self-contained. For BioASQ (Krithara et al., 2023), we focus on questions from Tasks 2B, 3B, and 4B, disregarding long answers to assess the

**Table 2:** Summary of Dataset statistics.

| Task | KG | $|\mathcal{V}|$ | $|\mathcal{E}|$ | Datasets | QA Type |
|---|---|---|---|---|---|
| Biomedical Reasoning | UMLS (Li et al., 2023) | 135 | 5,877 | PubmedQA (Jin et al., 2019)
BioASQ (Krithara et al., 2023)
ProcessBank (Berant et al., 2014) | Yes-No
Yes-No
Multiple-Choice |
| Commonsense Reasoning | 10% ConceptNet (Speer et al., 2018)
50% ConceptNet (Speer et al., 2018)
Full ConceptNet (Speer et al., 2018) | 223,863
607,483
844,158 | 208,510
1,042,550
2,085,099 | CommonsenseQA (Talmor et al., 2019) | Multiple-Choice |
| Opendomain Reasoning | 10% ConceptNet (Speer et al., 2018) | 223,863 | 208,510 | TruthfulQA (Lin et al., 2022) | Text Generation |

precision of the short responses each method provides. For Processbank (Berant et al., 2014), ground-truth annotations are **omitted**, providing only the question statement and choices. All three datasets are paired with a compact UMLS (Li et al., 2023) containing just 135 nodes. Regarding CommonsenseQA (Talmor et al., 2019), built with ConceptNet (Speer et al., 2018), we randomly sample ConceptNet subgraphs with varying edge ratios.

### 4.3. Competing baselines and backbone LLMs

We compare GIVE with I/O prompting (Brown et al., 2020), CoT prompting (Wei et al., 2023), text-based RAG (Lewis et al., 2021), ToG (Sun et al., 2024) and GraphRAG (Edge et al., 2024). Since GraphRAG is not suitable for large-scale KGs and struggles with context comprehension using smaller LLMs, it is excluded from certain comparisons. For biomedical reasoning tasks, we evaluate each method using GPT3.5-turbo, GPT-4, GPT4o-mini, and Llama3.1-70B-Instruct, aiming to demonstrate GIVE's ability to bridge the knowledge gap between smaller and larger LLMs. This is crucial in scenarios where acquiring knowledge from pretraining is challenging. For commonsense reasoning, we test GIVE on GPT3.5-turbo using ConceptNet with varying edge ratios, highlighting its robustness in managing both limited and noisy information. In open-domain reasoning, we showcase its effectiveness in general-domain QA tasks.

### 4.4. Biomedical Reasoning results

**GIVE enhances the effectiveness of smaller-sized LLMs beyond that of leading models.** Our initial observation reveals that, with minimal external knowledge, GIVE allows GPT3.5T to consistently outperform GPT4 on biomedical reasoning tasks, which are challenging due to the difficulty in training and accessing comprehensive knowledge bases. This challenge is particularly evident when comparing k-shot prompt results in various models, especially in tasks like PubmedQA (Jin et al., 2019) and ProcessBank (Berant et al., 2014). In these instances, GIVE effectively integrates training and inference knowledge using a sparse KG with just 135 nodes, incurring no additional training expense.

**GIVE is adaptable for LLMs of various sizes, effectively averting hallucination.** GIVE enhances the reasoning ability of LLMs of different sizes (GPT4 > GPT3.5T > Llama3.1 > GPT4o-mini). Notably, GIVE surpasses retrieval-based techniques that rely on limited external knowledge. Under our challenging experiment settings, triplets from DPR (Karpukhin et al., 2020) and ToG (Sun et al., 2024) didn't solve queries directly, the irrelevant information causes hallucinations. This occurs especially with strong models (GPT4/4o-mini). In this challenging scenario, GIVE consistently boosts their performance by properly alleviating hallucinations from irrelevant knowledge.

**Integrating affirmative, counterfactual and expert knowledge into increases the reliability of its reasoning processes.** The results by $\text{GIVE}_{a+c+e}$ utilizes the entirety of generated knowledge, as elaborated in Section 3.5, and delivers highly stable performance. This implies that the strategy for retrieving counterfactual knowledge, as discussed in Section 3.4.3, plays a crucial role in guiding the reasoning process in the specific-domain tasks. It also underscores the importance of effectively using retrieved expert knowledge that does not immediately resolve the query but contributes valuable context for more informed answer-generation.

### 4.5. Commonsense Reasoning Results

**GIVE demonstrates robustness and generalizability in reasoning with limited or noisy data.** As shown in Table 4, on the complete ConceptNet, GIVE increases accuracy by 3.4% and 4.9% over RAG (Lewis et al., 2021) and ToG (Sun et al., 2024). Retrieving information from a dense, domain-specific knowledge base is challenging due to numerous semantically-similar but irrelevant entities and triplets. Incorporating them directly in context easily causes hallucination. These findings confirm GIVE's robustness in generating useful information for prompting structured reasoning in LLMs using noisy external knowledge sources.

### 4.6. Open-domain Reasoning Results

**GIVE is effective in both domain-specific and open-domain reasoning tasks.** Alongside domain-specific rea-

**Table 3:** Performance comparison on biomedical QA. Retrieval-based methods are given access to a small UMLS (Li et al., 2023). We highlight the best performance gain of GIVE compared to different categories of competing methods.

| # Method/Dataset | GPT3.5-turbo | | | GPT4 | | | GPT4o-mini | | | Meta-Llama-3.1-70B-Instruct | | |
|---|---|---|---|---|---|---|---|---|---|---|---|---|
| | PubMedQA | BioASQ | ProcessBank | PubMedQA | BioASQ | ProcessBank | PubMedQA | BioASQ | ProcessBank | PubMedQA | BioASQ | ProcessBank |
| *Internal knowledge reasoning* | | | | | | | | | | | | |
| 1 I/O prompt | 46.2 | 43.5 | 67.3 | 42.2 | 88.2 | 64.8 | 23.4 | 88.7 | 79.4 | 48.0 | 91.0 | 85.4 |
| 2 CoT | 48.6 | 63.5 | 70.9 | 37.8 | 80.4 | 59.3 | 23.8 | 79.3 | 81.4 | 50.4 | 91.3 | 84.3 |
| *External knowledge (text) reasoning* | | | | | | | | | | | | |
| 3 RAG | 13.4 | 40.9 | 67.3 | 26.4 | 24.3 | 78.9 | 15.2 | 16.3 | 84.9 | 49.8 | 45.4 | 84.4 |
| *External knowledge (KG) reasoning* | | | | | | | | | | | | |
| 4 ToG | 17.6 | 18.0 | 66.8 | 19.1 | 15.4 | 81.4 | 21.8 | 10.1 | 84.4 | 38.4 | 31.0 | 85.9 |
| 5 GraphRAG | 23.4 | 10.3 | 71.3 | 26.5 | 11.2 | 80.9 | 22.6 | 10.1 | 84.9 | / | / | / |
| *Structured reasoning with internal and external knowledge(Our method)* | | | | | | | | | | | | |
| 5 $GIVE_a$ | 44.4 | 82.6 | 72.9 | 50.0 | **90.0** | 82.7 | 26.0 | **89.5** | 85.9 | 56.0 | 91.7 | 86.4 |
| 6 $GIVE_{a+c}$ | 49.8 | 86.1 | **73.9** | 50.2 | 80.6 | **83.3** | 27.4 | 81.9 | **87.4** | 56.2 | 91.7 | **86.9** |
| 7 $GIVE_{a+c+e}$ | **53.6** | **88.2** | 73.4 | 43.4 | 87.8 | 82.7 | 27.2 | 81.9 | 86.9 | 56.0 | **92.6** | 86.4 |
| 8 Best Gain(+%) | 7.4/40.2/36.0 | 44.7/47.3/77.9 | 6.6/6.6/7.1 | 12.4/23.8/31.1 | 9.6/65.7/78.8 | 24.0/4.4/2.4 | 4.0/12.2/5.6 | 10.2/73.2/79.4 | 8.0/2.5/3.0 | 8.2/6.4/17.8 | 1.6/47.2/61.6 | 2.6/2.5/1.0 |

**Table 4:** Performance comparison on CommonsenseQA, using GPT3.5-turbo. Retrieval-based methods are given access to a sub-graph of ConceptNet with different portions of randomly sampled triplets. We highlight the best performance gain of GIVE compared to different categories of competing methods.

| # Method / % of triplets | Commonsense QA | | |
|---|---|---|---|
| | 10% | 50% | 100% (Full) |
| *Internal knowledge reasoning* | | | |
| 1 I/O prompt | | 71.8 | |
| 2 CoT prompt | | 72.2 | |
| *External knowledge (text) reasoning* | | | |
| 3 RAG | 70.4 | 70.6 | 71.3 |
| *External knowledge (KG) reasoning* | | | |
| 4 ToG | 69.7 | 71.2 | 69.8 |
| *Internal and external knowledge reasoning* | | | |
| 5 $GIVE_a$ | 73.3 | 73.6 | 74.2 |
| 6 $GIVE_{a+c}$ | 73.4 | 73.6 | 74.2 |
| 7 $GIVE_{a+c+e}$ | **73.5** | **73.8** | **74.7** |
| 8 Best Gain(+%) | 1.7/3.1/3.8 | 2.0/3.2/2.6 | 2.9/3.4/4.9 |

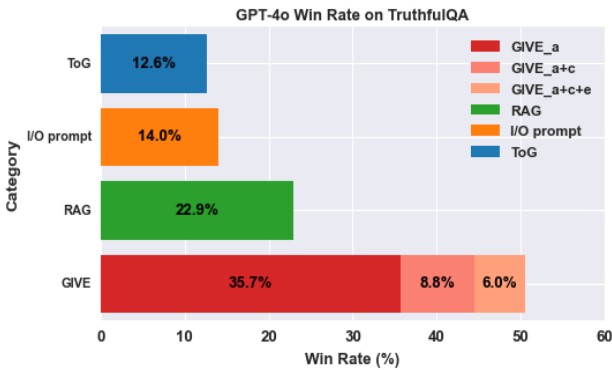

**Figure 3:** GPT4o win rate on TruthfulQA (Lin et al., 2022) in %.

It was observed that $GIVE_a$ yields the most accurate answers compared to $GIVE_{a+c}$ and $GIVE_{a+c+e}$, as the dataset focuses on information veracity. For example, "What is the original meaning of SOS?" Here, counter-factual knowledge is not useful. Humans don't answer by considering "SOS does not mean..." statements, which could lead to hallucinations in LLMs. Therefore, in open-domain datasets, it is advisable to prioritize affirmative knowledge and $GIVE_a$ answers.

## 4.7. Ablation Study

### 4.7.1. NUMBER OF ADDITIONAL ENTITIES

**GIVE demonstrates optimal performance with a minimal number of additional entities to conduct divergent reasoning.** The pivotal parameter for GIVE is the count of supplementary entities added to each concept group. We examine its impact on GIVE's performance through experiments in biomedical reasoning employing GPT3.5-turbo, with KG entity numbers ranging from 0 to 3. As depicted in Figure 4, GIVE's performance initially enhances with an increase in KG entities per group from 0 to 2, yet declines when increased to 3. This pattern is consistent across all

soning tasks, we conduct supplementary experiments to evaluate GIVE's performance on open-domain QA dataset. We assess each method's performance on the TruthfulQA (Lin et al., 2022) answer generation task. For retrieval-based methods (RAG, ToG, GIVE), we employ a 10% ConceptNet ratio to test their capability in generating open-domain answers with limited information. We utilize GPT4o to evaluate the win rate, as presented in Figure 3. For each question, GPT4o receives the best answer and each method's response, scores them based on their semantic similarity to the ground truth, and selects the highest scoring answer as the winner. As indicated in Figure 3, GIVE attains the highest score in 50.3% of the TruthfulQA questions (Lin et al., 2022), demonstrating its efficacy in both scientific reasoning and open-domain question answering.

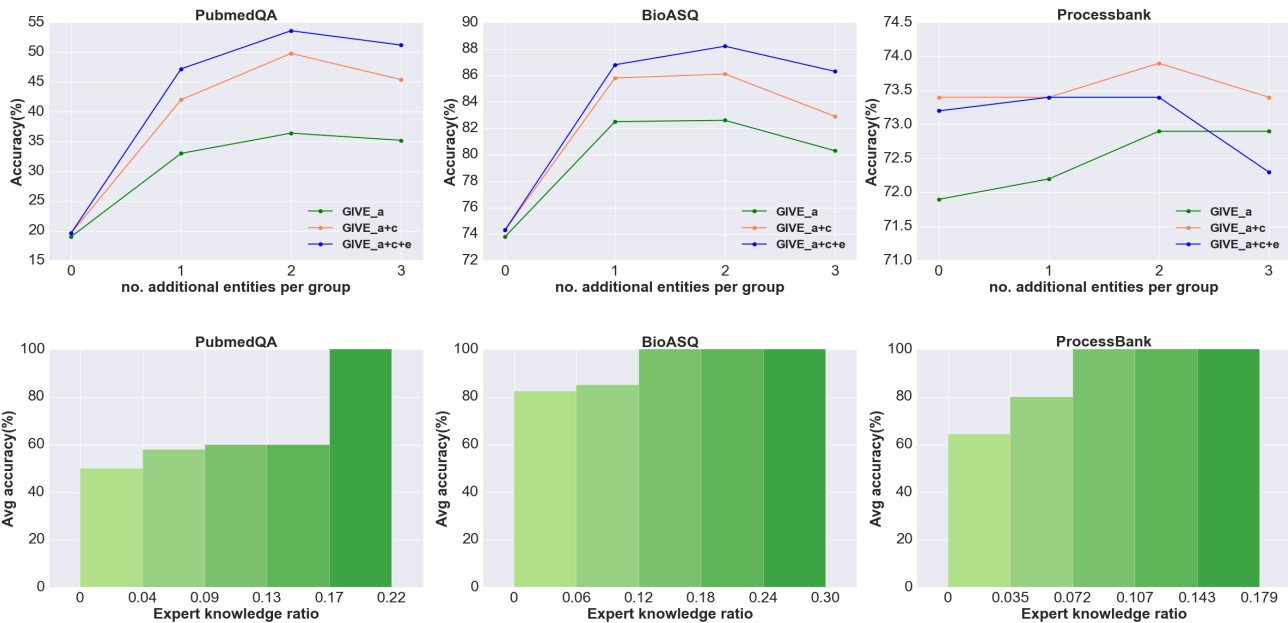

**Figure 4:** Different factors that may impact GIVE's performance: **(upper)** GIVE's performance VS no.additional entities per group. **(lower)** GIVE's performance VS expert knowledge ratio of the "veracity extrapolation" process

datasets. The noticeable improvement in performance with an increase in additional concepts from 0 to 1 underscores the success of our "Graph Inspired Veracity Extrapolation" method, as detailed in Section 3.4.3. This suggests that LLMs' potential for divergent thinking has been underestimated, due to the emphasis on retrieving precise knowledge.

**Table 5:** Performance (Accuracy %) comparison of GIVE using only inner-group connections, inter-group connections, both inner/inter group connections on 100 randomly selected questions from each dataset.

| Dataset/Knowledge | Inner-Group only | Inter-Group only | Inner & Inter Group |
|---|---|---|---|
| PubmedQA | 14 | 52 | 56 |
| BioASQ | 50 | 84 | 88 |
| ProcessBank | 70 | 70 | 76 |
| CommonsenseQA | 66 | 74 | 76 |

### 4.7.2. INNER-GROUP AND INTER-GROUP CONNECTIONS

We conduct experiments to test the importance of each type of connections introduced by GIVE. The results are presented in Table 5. We conclude that both inner-group and inter-group connections are necessary, but inter-group connections contribute the most to the performance. This is because the inner-group connections are added to bridge the query and the knowledge graph concepts, and it's purely built on the model's internal knowledge, so they are easier to be automatically inferred by the model. The inter-group connections produce the necessary knowledge to bridge the different entity groups thus prompt a faithful reasoning process to solve the query, which are hard to reason

from the parametric knowledge, and need to be guided by frameworks like GIVE.

### 4.7.3. WHAT MAKES GOOD "INSPIRATIONS"?

To further scrutinize the factors leading to GIVE's high performance, we classify the dataset questions by "expert knowledge ratio" in GIVE's "veracity extrapolation" process on it. For each question, we compute the proportion of expert knowledge in the extrapolated knowledge set. Specifically, for query $x$, the expert ratio is defined as $\frac{|\tilde{\mathcal{R}}^e(x)|}{|\tilde{\mathcal{R}}^a(x)|+|\tilde{\mathcal{R}}^e(x)|+|\tilde{\mathcal{R}}^c(x)|}$, where $\tilde{\mathcal{R}}^e(x)$, $\tilde{\mathcal{R}}^a(x)$, and $\tilde{\mathcal{R}}^c(x)$ represent sets of expert, affirmative, and counter-factual knowledge retrieved from Section 3.4.3. We assess GIVE's average accuracy across questions with varying expert ratios and display findings in Figure 4. We observe a positive correlation between GIVE's performance and the expert ratio. This occurs because more expert information provides GIVE with concrete candidate relations and entities for LLM to perform divergent thinking during the "veracity extrapolation" process. Limited expert information leaves no "inspiration" for divergent thinking, the final knowledge set consists of only the inner-group connections and the open-relation inter-group connections, detailed in Section 3.4. GIVE uses ground truth knowledge as a qualitative "supervision" to guide the model on possible entity relationships, supporting our premise that both external and internal knowledge are insufficient alone for knowledge-intensive scientific tasks, prompting the design of GIVE to bridge this gap.

# 5. Conclusion

We introduce Graph Inspired Veracity Extrapolation (GIVE), a structured reasoning framework to enhance LLM reasoning in scientific fields. This is achieved by leveraging divergent thinking guided by limited expert input. Rather than focusing on explicit information retrieval or relying solely on the parametric knowledge, GIVE adopts an approach akin to human cognitive reasoning process, by combining high-level expert information with the model's internal knowledge through "veracity extrapolation". This bridges the limited expert knowledge and queries, thus significantly boosts performance in both domain-specific and open-domain reasoning tasks, demonstrating robustness with limited and noisy knowledge bases. GIVE addresses hallucination issues in retrieval-dependent methods on incomplete knowledge sources, at the same time, adding information for more comprehensive answers compared to reasoning methods that solely rely on internal knowledge. Our study highlights the great potential of LLMs to perform divergent reasoning with minimal external guidance. Further research is needed on leveraging external knowledge as a catalyst to "inspire" LLMs to reason, rather than providing a comprehensive "long-answer" style context. More generalizable RL post-training strategies are also needed, on tasks where additional information is needed to generate the correct logic chain. GIVE holds particular relevance for deploying LLMs in scientific fields where exhaustive training and retrieval of knowledge are impractical. **Intelligent agents draw inspiration from external cues to perform accurate reasoning, not thinking hard by itself or just echo the context.**

## Impact Statement

This paper presents work whose goal is to advance the field of Machine Learning. There are many potential societal consequences of our work, none which we feel must be specifically highlighted here.

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

# A. Algorithm for GIVE

We offer a comprehensive outline of the entire GIVE procedure and explain its detailed algorithm in Algorithm 1

---

**Algorithm 1** GIVE

---

**Input:** Entity groups $\mathbb{N}_x = \{N_x^i\}_{i=1}^k$; Possible relations between two entity groups $\mathbb{R}_x = \{R_x^{ij}\}_{i,j=1}^k$; Knowledge Graph G

**Output:** $\tilde{\mathcal{R}}(x)$, the approximated gold knowledge set that helps to solve query x

$\tilde{\mathcal{R}}^{\text{a}}(x) \leftarrow \varnothing$

$\tilde{\mathcal{R}}^{\text{c}}(x) \leftarrow \varnothing$

$\tilde{\mathcal{R}}^{\text{e}}(x) \leftarrow \varnothing$

**for** all queried entity $e_x^i$ and their relevant concepts $y_x^j \in N_x^i$ **do**

  /* build inner-group connections */

  $(e_x^i, \_, y_x^j) \rightarrow \text{LLM} \rightarrow (e_x^i, r_x^{ij}, y_x^j)$

  $\tilde{\mathcal{R}}^{\text{a}}(x) \leftarrow \tilde{\mathcal{R}}^{\text{a}}(x) \cup \{(e_x^i, r_x^{ij}, y_x^j)\}$

**end for**

**for** $(N_x^i, N_x^j)$ pair in $\mathbb{N}_x \times \mathbb{N}_x$ **do**

  /* build inter-group connections */

  retrieve all triplets $\tilde{\mathcal{R}}_{ij}^{\text{e}}(x) \in \mathcal{E}_G$ connecting any node in $N_x^i$ and any node in $N_x^j$

  $\tilde{\mathcal{R}}^{\text{e}}(x) = \tilde{\mathcal{R}}^{\text{e}}(x) \cup \tilde{\mathcal{R}}_{ij}^{\text{e}}(x)$

  **for** all triplets $(n_x^i, r_x^{ij}, n_x^j)$ in $(N_x^i \times R_x^{ij} \times N_x^j)$ **do**

    $(n_x^i, r_x^{ij}, n_x^j) \rightarrow \text{LLM} \rightarrow \{\text{yes, no, maybe}\}$

    **if** yes **then**

      $\tilde{\mathcal{R}}^{\text{a}}(x) \leftarrow \tilde{\mathcal{R}}^{\text{a}}(x) \cup \{(n_x^i, r_x^{ij}, n_x^j)\}$

    **end if**

    **if** no **then**

      $\tilde{\mathcal{R}}^{\text{c}}(x) \leftarrow \tilde{\mathcal{R}}^{\text{c}}(x) \cup \{(n_x^i, \text{not } r_x^{ij}, n_x^j)\}$

    **end if**

  **end for**

**end for**

**return** $\tilde{\mathcal{R}}^{\text{a}}(x) \cup \tilde{\mathcal{R}}^{\text{c}}(x) \cup \tilde{\mathcal{R}}^{\text{e}}(x)$

---

# B. Additional Ablation studies

In addition to Section 4.7, we conduct more detailed ablation studies for GIVE to study the robustness of the proposed method and other factors that may influence its performance. All experiments in this Section are based on 50 randomly generated examples for each dataset.

## B.1. Different ways of prompting

We perform additional experiments to study how different prompting strategies influence the performance of GIVE. We verbalize the retrieved knowledge and prompt them in the form of triplets and text, the results are presented in Table 6. We notice that in most cases, prompting the knowledge in triplets yields to higher accuracy than prompting knowledge in text. This is because the structure of triplets naturally provides an easier way for the LLM to connect the related entities and build faithful logical chain to solve the question. However, for text-based information, additional analyzing step is needed to understand the text before it links the useful information together, which is a difficult task for reasoning-intensive queries where the volume of additional knowledge is high.

**Table 6:** Performance of GIVE using different prompting methods on 50 randomly chosen examples for each dataset. We highlight in green the better-performed prompting method and the performance difference.

| # Prompting Method / dataset | GPT3.5-turbo | | | |
|---|---|---|---|---|
| | PubmedQA | BioASQ | Processbank | CSQA |
| **GIVE$_{\text{a}}$** | | | | |
| 1 Triplet prompt | 32 | 86 | 76 | 74 |
| 2 Text prompt | 46 | 86 | 74 | 62 |
| **GIVE$_{\text{a+c}}$** | | | | |
| 1 Triplet prompt | 56 | 86 | 74 | 76 |
| 2 Text prompt | 54 | 84 | 74 | 70 |
| **GIVE$_{\text{a+c+e}}$** | | | | |
| 1 Triplet prompt | 52 | 88 | 70 | 76 |
| 2 Text prompt | 54 | 84 | 72 | 68 |

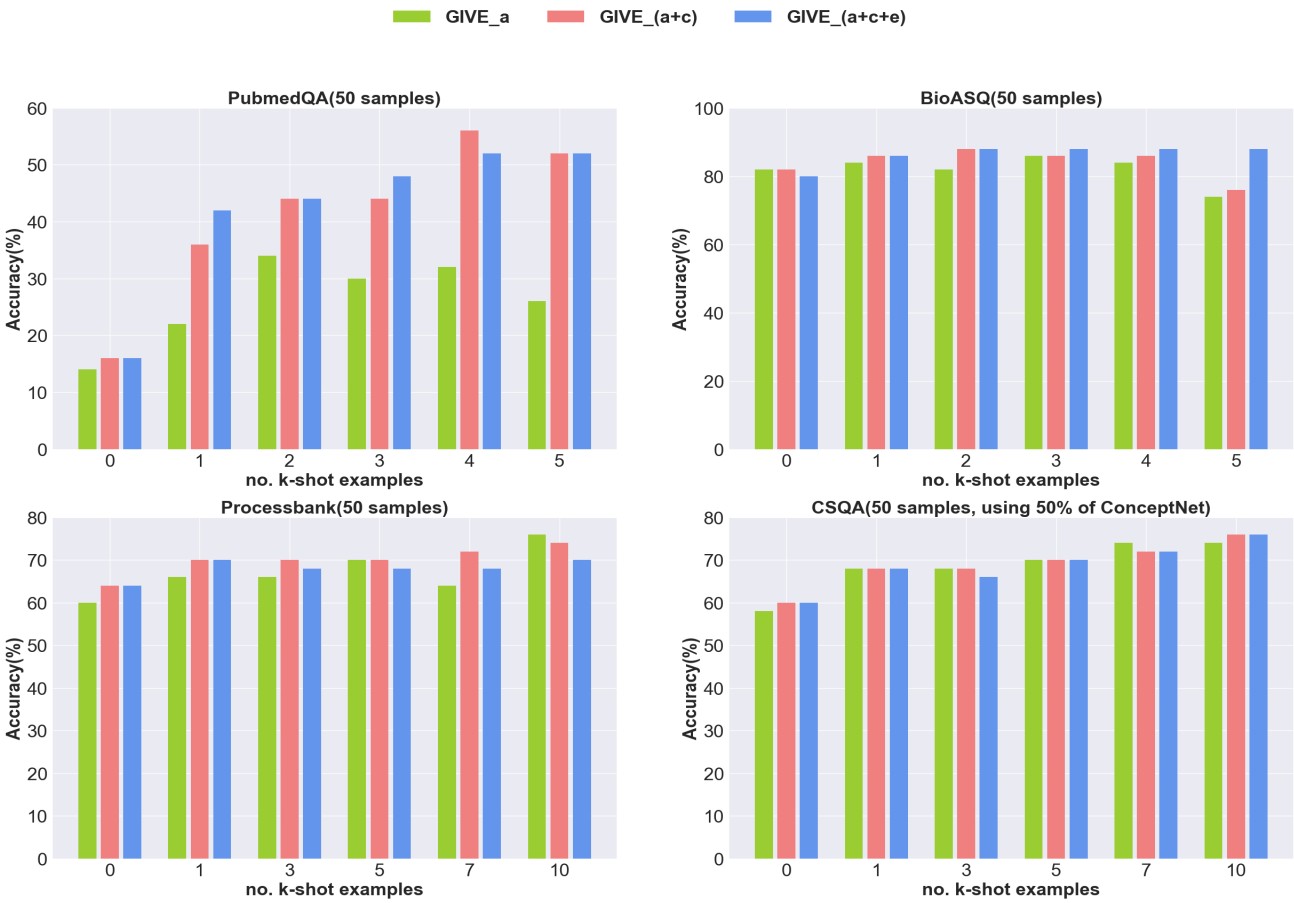

**Figure 5:** Sensitivity analysis of GIVE on different number of seeded examples.

## B.2. Number of seeded examples

To better understand how difficult it is for LLMs to get the generalized ability to adopt the knowledge generated by GIVE to build the structured reasoning chain, we study the performance of GIVE by providing different number of examples in the prompt. For yes-no datasets PubmedQA (Jin et al., 2019) and BioASQ (Krithara et al., 2023), we randomly choose k of {0, 1, 2, 3, 4, 5} examples. For multiple-choice datasets Processbank (Berant et al., 2014) and CSQA (Talmor et al., 2019), we choose k of {0, 1, 3, 5, 7, 10}. The results are presented in Figure 5.

We observe that although the performance of GIVE increases as we give more seeded examples in the prompt, the only one large performance upgrade happens when we increase the number of examples from 0 to 1. This implies that GIVE is a generalizable framework for the LLM to easily adopt. The high performance of GIVE does not rely on large number of examples, but stems from the high quality of the synthetic data it generates.

**Table 7:** Performance of GIVE using GPT3.5-turbo and encoding SentenceTransformers of different sizes to search for relevant entities to build entity group (Section 3.2). Results are based on 50 randomly generated samples for each dataset. We highlight the results from the best performing model in green.

| # | Encoding model(size) / dataset | PubmedQA | BioASQ | Processbank | CSQA |
|---|---|---|---|---|---|
| | | \multicolumn{4}{c}{GPT3.5-turbo} | | | |
| | | \multicolumn{4}{c}{**GIVE$_a$**} | | | |
| 1 | paraphrase-albert-small-v2(43M) | 44 | 84 | 74 | 68 |
| 2 | all-MiniLM-L6-v2(80M) | 32 | 86 | 76 | 74 |
| 3 | all-MiniLM-L12-v2(120M) | 24 | 80 | 62 | 72 |
| 4 | all-mpnet-base-v2(420M) | 38 | 88 | 66 | 64 |
| | | \multicolumn{4}{c}{**GIVE$_{a+c}$**} | | | |
| 1 | paraphrase-albert-small-v2(43M) | 54 | 82 | 76 | 70 |
| 2 | all-MiniLM-L6-v2(80M) | 56 | 86 | 74 | 76 |
| 3 | all-MiniLM-L12-v2(120M) | 52 | 82 | 62 | 70 |
| 4 | all-mpnet-base-v2(420M) | 52 | 86 | 62 | 64 |
| | | \multicolumn{4}{c}{**GIVE$_{a+c+e}$**} | | | |
| 1 | paraphrase-albert-small-v2(43M) | 52 | 84 | 76 | 72 |
| 2 | all-MiniLM-L6-v2(80M) | 52 | 88 | 70 | 76 |
| 3 | all-MiniLM-L12-v2(120M) | 54 | 82 | 62 | 70 |
| 4 | all-mpnet-base-v2(420M) | 52 | 88 | 60 | 64 |

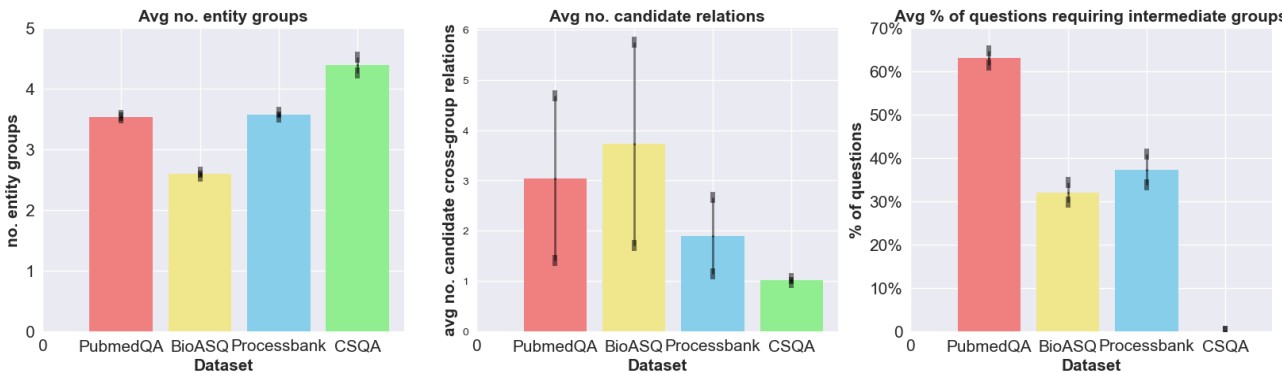

**Figure 6:** Average number of entity groups (left), average number of candidate relations between two groups (middle) and average percentage of questions that requires intermediate entity group (right) for each dataset included in Section 4.4 and 4.5 for 5 runs. For CSQA, we report the results on 50% triplets version of ConceptNet.

## B.3. Encoding model size

In Section 3.2, we employ SentenceTransformer as encoder model to measure the text similarities for entity group construction. We investigate the impacts of using different sizes on the performance of GIVE, and demonstrate the results in Table 7. We see that although larger size encoder models achieve better sentence embedding or performance semantic search performance, small to middle size encoders tend to perform more consistently on all datasets. For the best-performing GIVE$_{a+c+e}$, the 80M encoder (all-MiniLM-L6-v2) achieves 8% higher accuracy than the 420M one (all-mpnet-base-v2). The results show that larger size encoders do not necessarily better measure text similarity between specific domain terms. On the other hand, the performance of GIVE does not rely on the size of the models employed, which enhances the efficiency of GIVE.

## C. Detailed analysis of GIVE

### C.1. Efficiency of GIVE

In Section 3.6 we discussed the efficiency of GIVE and concluded that the key factors that influence the number of LLM calls required by GIVE is the number of entity groups detected for the query and the number of candidate relations between each pair of entity groups. To conduct a more detailed study on the scale of them, we run GIVE 5 times on 50 randomly selected questions for each of the datasets we included in Section 4.4 and 4.5, and we report the average number of entity groups, average number of candidate relations to connect two entity groups, and average percentage of questions that requires intermediate entity groups (3.4.2) for multi-step reasoning. The results are presented in Figure 6.

We observe that on average, GIVE requires around 3 entities groups for each question in the Biomedical datasets (PubmedQA, BioASQ, Processbank), between each datasets, there could be 1 to 6 candidate relations. For commonsenseQA, 4 entity groups on average are detected because the dataset has 5 candidate options, between each pair of entity groups, only 1 candidate relation is detected in general. We also notice that 60% of the questions in PubMedQA requires intermediate group. That is the reason why PubMedQA tends to need more entity groups than BioASQ as a "yes-no" QA dataset. This implies one of the potential method to improve efficiency of GIVE is to disable intermediate group detection. On the

Table 8: Token consumption comparison between GIVE and ToG

| Dataset | Avg Input Tokens | | Avg Output Tokens | |
|---|---|---|---|---|
| | GIVE$_{n=1}$ | ToG | GIVE$_{n=1}$ | ToG |
| PubmedQA | 14518.5 | 12701.1 | 183.1 | 104.2 |
| BioASQ | 7970.3 | 7010.0 | 80.0 | 60.2 |
| ProcessBank | 19460.5 | 11995.6 | 232.5 | 100.5 |
| CSQA/10% ConceptNet | 5321.1 | 4934.8 | 34.9 | 23.1 |
| CSQA/50% ConceptNet | 7203.0 | 6704.1 | 45.2 | 34.8 |
| CSQA/100% ConceptNet | 7398.7 | 6679.7 | 46.3 | 36.6 |

other hand, we can use the LLM to prune the candidate connections in batches, which means in Section 3.4.3, instead of asking LLM "yes" or "no" for each potential connection, we can prompt the LLM with a set k of relations and let it select out which ones are true of false, which will devide the total number of LLM calls by the factor of k for GIVE.

We further conduct experiments to compare the efficiency of GIVE with RAG (Lewis et al., 2021) and ToG (Sun et al., 2024), in terms of running time and context length, for every experiment setting we include in Section 4, the results are presented in Table 9. ALso, to conduct an in-depth analysis of token consumption, we compare the average number of input.output tokens for GIVE and ToG (Sun et al., 2024), as in Table 8.

**The computational cost of GIVE remains reasonable as we increase the size and density of the KG.** (1) In terms of running time, when we increase the density (number of edges) to ×5 or ×10 on ConceptNet (Speer et al., 2018), we see a sub-linear running time increase for GIVE. Even with n=2, GIVE achieves shorter or comparable running time with ToG (Sun et al., 2024). This proves the $\mathcal{O}(rm^2n^2)$ running time of GIVE, as discussed in Section 3.4.3. When we increase the density of the KG, the only factor that will change is $r$, which is the number of relations between two entity groups, the increase of which is strictly upper-bounded by the increase of total number of edges in KG. Besides, the running time of GIVE is independent to the size of the KG, we see this if we compare its running time on small UMLS of 135 nodes and the ConceptNets which have hundreds of thousands of entities, because GIVE always selects the most important entities related to the query to induce knowledge, and the cost of the entity selection phase is very low if we pre-compute the embeddings. (2) In terms of context length, GIVE does not suffer from overwhelming long context on large or dense KGs. In fact, the context length of GIVE is also largely decided by the number of relations between two groups. That is why we see that on PubmedQA where the cross-group KG knowledge is rich, GIVE can induce large number of affirmative and counterfactual knowledge, thus provides the biggest performance increase compared to RAG or ToG. It is also worth noticing the recent LLMs are making fast progress in overcoming the limitation of input length. For example, Llama 3.1 series(OpenAI et al., 2024b) support up to 128k tokens, compared to Llama 3 which supports only 8,192 tokens. For GPT series models, the maximum context window size also grows from 4.1k tokens (which translates to around 3k words) of GPT3.5-turbo to 32k of GPT4 (Chowdhery et al., 2022) and GPT4o. Such

**Table 9:** Efficiency comparison between GIVE and RAG (Lewis et al., 2021), ToG (Sun et al., 2024) on 100 random questions. We report the average running time in seconds, context length in no. words, and accuracy in %. For RAG, we retrieved top 10 knowledge and for ToG, we use search depth=5 to maximize their performance. For GIVE, we report the results for both n=1 and n=2, where n is the number of additional KG concepts per group. $\tilde{\mathcal{R}}^a(x)$, $\tilde{\mathcal{R}}^c(x)$, $\tilde{\mathcal{R}}^e(x)$ are the retrieved affirmative knowledge set, counter-factual knowledge set and expert KG knowledge set.

| # | Method/Dataset | time(s) | Context Length (# words) $\tilde{\mathcal{T}}_x^a(G)$ | $\tilde{\mathcal{T}}_x^c(G)$ | $\tilde{\mathcal{T}}_x^e(G)$ | Acc(%) GIVE$_a$ | GIVE$_{a+c}$ | GIVE$_{a+c+e}$ |
|---|---|---|---|---|---|---|---|---|
| | | | | | **PubmedQA on UMLS** | | | |
| 1 | RAG | 2.7 | | 64.7 | | | 14 | |
| 2 | ToG | 15.9 | | 73.8 | | | 16 | |
| 3 | GIVE$_{n=1}$ | 33.6 | 192.5 | 105.9 | 16.1 | 35 | 45 | 45 |
| 4 | GIVE$_{n=2}$ | 103.8 | 456.3 | 486.1 | 57.6 | 41 | **48** | **48** |
| | | | | | **BioASQ on UMLS** | | | |
| 1 | RAG | 2.8 | | 66.5 | | | 43 | |
| 2 | ToG | 10.3 | | 42.7 | | | 17 | |
| 3 | GIVE$_{n=1}$ | 15.3 | 83.6 | 33.8 | 9.0 | 79 | 81 | 83 |
| 4 | GIVE$_{n=2}$ | 45.3 | 205.5 | 176.5 | 32.1 | 80 | 84 | **90** |
| | | | | | **Processbank on UMLS** | | | |
| 1 | RAG | 2.8 | | 61.7 | | | 68 | |
| 2 | ToG | 15.6 | | 54.2 | | | 60 | |
| 3 | GIVE$_{n=1}$ | 35.2 | 151.2 | 226.9 | 7.5 | 67 | 68 | 68 |
| 4 | GIVE$_{n=2}$ | 93.8 | 354.0 | 649.5 | 28.9 | 71 | 71 | **72** |
| | | | | | **CSQA on 10% ConceptNet** | | | |
| 1 | RAG | 0.6 | | 30 | | | 64 | |
| 2 | ToG | 39.3 | | 106.6 | | | 67 | |
| 3 | GIVE$_{n=1}$ | 26.5 | 41.1 | 38.4 | 0.1 | 70 | **71** | 71 |
| 4 | GIVE$_{n=2}$ | 36.3 | 93.2 | 83.3 | 0.2 | 70 | 70 | 68 |
| | | | | | **CSQA on 50% ConceptNet** | | | |
| 1 | RAG | 1.1 | | 30 | | | 66 | |
| 2 | ToG | 102.2 | | 217.0 | | | 67 | |
| 3 | GIVE$_{n=1}$ | 74.0 | 39.9 | 43.5 | 0.2 | 69 | 64 | 65 |
| 4 | GIVE$_{n=2}$ | 82.0 | 86.9 | 91.6 | 0.5 | 73 | **76** | 75 |
| | | | | | **CSQA on full ConceptNet** | | | |
| 1 | RAG | 1.7 | | 30 | | | 69 | |
| 2 | ToG | 125.2 | | 213.7 | | | 63 | |
| 3 | GIVE$_{n=1}$ | 124.2 | 41.8 | 45.9 | 0.5 | 67 | 69 | 68 |
| 4 | GIVE$_{n=2}$ | 129.9 | 83.4 | 89.9 | 0.6 | 72 | **77** | **77** |

progress makes scaling inference time compute techniques like GIVE much more applicable, and we expect even large progress in maximum tokens on further models. GIVE is far from reaching such context length limitations according to Table 9. There are also concrete solutions to easily further reduce both running time context length of GIVE: When building the knowledge sets (Section 3.4.3), we can apply a divide-and-conquer manner to prune the knowledge in batches. When generating answers, we can apply similar techniques in GraphRAG (Edge et al., 2024), to use an additional agent to summarize the retrieved knowledge sets into shorter paragraphs before feeding to the answer generator.

**GIVE$_{n=1}$ provides a good trade-off between accuracy and efficiency.** Although the hyper-parameter n=2 yields to the best accuracy in most scenarios, we see that even when we use only one additional KG entity per group, GIVE achieves better or at least the same accuracy, compared to ToG and RAG. These results further emphasize the importance of the proposed framework to incorporate structured information during inference time reasoning, at the same time, provide the practicer with a balanced alternative to use n=1 with limited compute resource, but at the same time achieve good performance.

**GIVE is able to generate high quality synthetic data using very limited external knowledge.** If we compare the accuracy

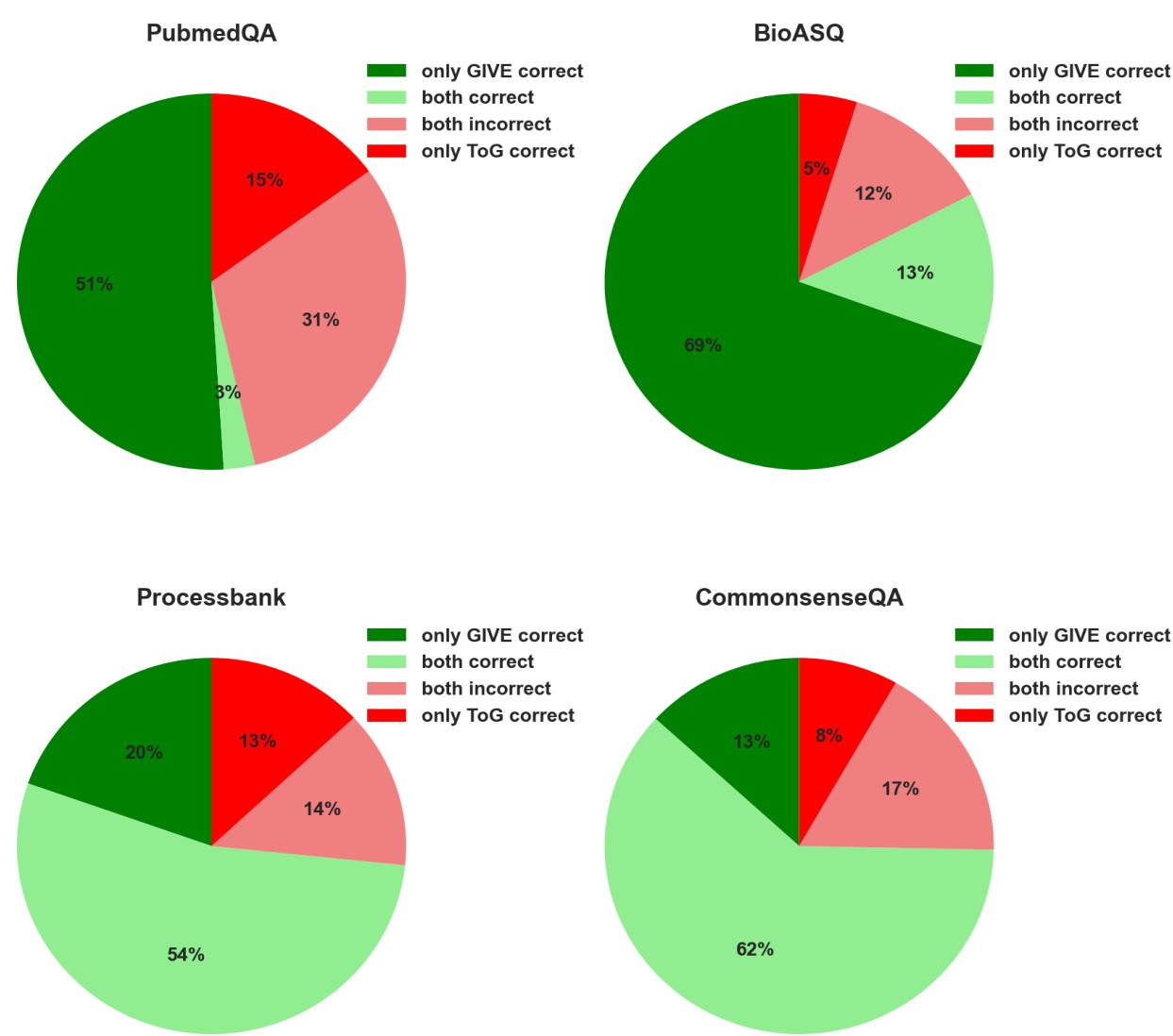

**Figure 7:** The proportions of questions answered correctly by GIVE and ToG, on PubmedQA, BioASQ, Processbank and CommonsenseQA

increase offered by GIVE and the context length, we see a positive correlation between them. Related discussion is also included in the previous subsection that the performance of GIVE is related to the expert knowledge ratio and the number of retrieved knowledge. The results further proved that the generated knowledge is of very high-quality. As a result, GIVE has great potential to serve as a synthetic data generating algorithm in other fine-tuning tasks, such as RLHF or RL for reasoning.

### C.2. Detailed comparison with retrieval-based methods

In addition to Table 3, we conduct detailed performance comparison against text-based reasoning method RAG (Lewis et al., 2021) and KG based reasoning method (Sun et al., 2024), we calculate the portions of questions answered correctly by each method and present the statistics in Figure 7 and Figure 8.

We observe that on three of the four datasets (BioASQ, Processbank, CommonsenseQA), we included in our experiments, most of the questions answered correctly by ToG or RAG is also answered correctly by GIVE. We see this by calculating the ratio $\frac{\text{only ToG/RAG correct}}{\text{only ToG/RAG correct+both correct}}$. For example, its 11% on CommonsenseQA for ToG, meaning that 89% of the questions it answered correctly is also answered correctly by GIVE. On PubmedQA, this ratio is large because RAG and ToG both get

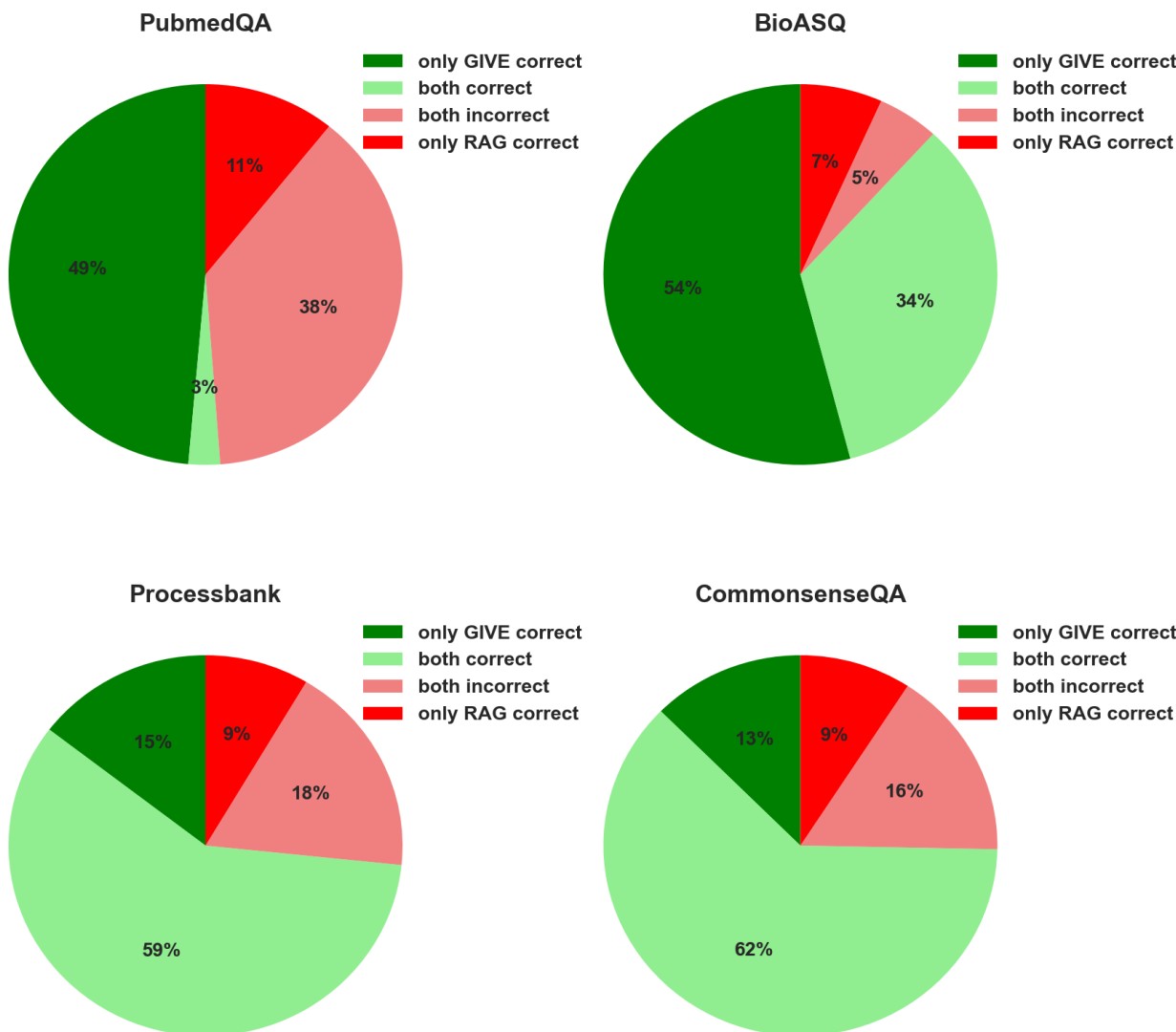

**Figure 8:** The proportions of questions answered correctly by GIVE and RAG, on PubmedQA, BioASQ, Processbank and CommonsenseQA

very poor performance, and for most questions, GIVE is the only method that can derive the correct answer. The results further prove that only very few questions in these scientific-domain datasets can be directly answered by the knowledge contained in the sparse KG, this further highlights the importance of the proposed "Veracity Extrapolation" process to combine internal knowledge and external knowledge to solve challenging scientific questions.

## D. Prompts and Example Responses

### D.1. IO Prompt

> You are a helpful assistant that answers a given question about medical knowledge with yes, no or maybe, based on your own knowledge.
> [k-shot EXAMPLES]
> Q: Traumatic aortic injury: does the anatomy of the aortic arch influence aortic trauma severity?
> **Output: no**

### D.2. CoT Prompt

> You are a helpful assistant that answers a given question about medical knowledge with yes, no or maybe, based on your own knowledge.
> [k-shot EXAMPLES]
> Q: Traumatic aortic injury: does the anatomy of the aortic arch influence aortic trauma severity? **Let's think step by step.**
> **Output: maybe**

### D.3. RAG Prompt

For RAG, we provide both the correct textual knowledge and reasoning chain for each of the k-shot examples.

> You are a helpful assistant that answers a given question about medical knowledge with yes, no or maybe, based on the retrieved textual knowledge "entity relation entity" from an expert knowledge graph.
> [k-shot EXAMPLES]
> Q: Traumatic aortic injury: does the anatomy of the aortic arch influence aortic trauma severity?
> Knowledge: [Textual knowledge]
> **Output: no**

### D.4. ToG Prompt

We follow the official implementation of ToG (**Sun et al., 2024**) and use the default prompts. We replace the k-shot examples to be examples randomly selected for each dataset, and we provide the correct reasoning chain. Overall, we use exact the same k-shot examples for ToG and our method to guarantee fair comparison.

Exemplar prompt for retrieving top entities:

> Please retrieve the top entities (separated by semicolon) that contribute to the question.
> [EXAMPLES]
> Q: Traumatic aortic injury: does the anatomy of the aortic arch influence aortic trauma severity?
> **Output: [Entities retrieved]**

Exemplar prompt for pruning relations:

> Please retrieve 1 relation that contributes to the question the most from the given relation list. The answer must be one of the given relations.
> [EXAMPLES]
> Q: Traumatic aortic injury: does the anatomy of the aortic arch influence aortic trauma severity?
> Relations: [Relations list]
> **Output: [Relationship selected]**

Exemplar prompt for pruning entities:

> Please score the entities' contribution to the question on a scale from 0 to 1 (the sum of the scores of all entities is 1).
> [EXAMPLES]
> Q: Traumatic aortic injury: does the anatomy of the aortic arch influence aortic trauma severity?
> Relation: [Relationship selected]
> Entities: [Entities list]
> **Output: [Entity selected]**

Exemplar prompt for evaluating knowledge sufficiency:

> Given a question and the associated retrieved knowledge graph triplets (entity, relation, entity), you are asked to answer whether it's sufficient for you to answer the question with these triplets and your knowledge (yes or no).
> [EXAMPLES]
> Q: Traumatic aortic injury: does the anatomy of the aortic arch influence aortic trauma severity?
> Knowledge triplets: [currently retrieved knowledge triplets]
> **Output: [yes/no]**

Exemplar prompt for ToG answering the question:

> Given a question and the associated retrieved knowledge graph triplets (entity, relation, entity), you are asked to answer the question with these triplets and your knowledge.
> [k-shot EXAMPLES]
> Q: Traumatic aortic injury: does the anatomy of the aortic arch influence aortic trauma severity?
> Knowledge triplets: [retrieved knowledge triplets]
> **Output: maybe**

## D.5. GraphRAG prompt

We follow the networkx implementation of GraphRAG (**Edge et al., 2024**) and use the default prompts. We replace the k-shot examples to be examples randomly selected for each dataset, and we provide the correct reasoning chain. The k-shot examples are provided during the intermediate answers generating step. Overall, we use exact the same k-shot examples for GraphRAG and our method to guarantee fair comparison.

Exemplar prompt for summarizing each detected community:

> Summarize the following community of entities and relationships.
> [Description of communities]
> **Output: [List of summaries of each community group]**

Exemplar prompt for generating intermediate answers from community summaries:

You are a helpful assistant that answers a given biomedical question based on the provided summary. You can find some examples below: + [k-shot examples]
Query: [Question]
Summary: [Summary list]
**Output: [List of intermediate answers]**

Exemplar prompt for combining intermediate answers into a final answer:

You are a helpful assistant that answers a biomedical question with yes, no or maybe, based on some intermediate answers.
Query: [Question]
Intermediate answers: [List of intermediate answers]
**Output: [yes/no/maybe]**

## D.6. GIVE Prompt

Exemplar prompt for extracting and ranking the entities in the question:

Please retrieve the top entities that contribute to the question. Answer only the top entities, separated by comma.
[EXAMPLES]
Question: Traumatic aortic injury: does the anatomy of the aortic arch influence aortic trauma severity?
**Output: ['traumatic aortic injury', 'anatomy', 'aortic arch', 'aortic trauma severity']**

Exemplar prompt for extracting the relationships in the question:

Please retrieve the relationships that connect the given entities in the question.
[EXAMPLES]
Question: Traumatic aortic injury: does the anatomy of the aortic arch influence aortic trauma severity?
Entities: traumatic aortic injury, anatomy, aortic arch, aortic trauma severity
**Output: ['influence']**

Exemplar prompt for generating relationships between two given entities:

You are a helpful assistant that answers a short relationship in a few words between two given biomedical entities.
[EXAMPLES]
Entities: traumatic aortic injury, injury and poisoning
**Output: "is a"**

Exemplar prompt for determining if relations exists between cross group entities:

You are a helpful assistant that answers yes, no or maybe depending on the correctness of the given statement.
Injury or poisoning is the result of organism function. Is it true?
**Output: "No"**

Exemplar prompt for selecting optimal 2-hop path for intermediate entity group construction:

You are a helpful assistant that selects one from the given knowledge facts (entity, relation, entity, relation, entity), that is most important to the given question.
Knowledge Facts:
(steroid, affects, organ or tissue function, affects, invertebrate),
(steroid, affects, experimental model of disease, manifestation of, injury or poisoning),
(anatomical abnormality, manifestation of, organism function, affects, clinical attribute)...
Question to answer: Traumatic aortic injury: does the anatomy of the aortic arch influence aortic trauma severity?
**Output: (anatomical abnormality, manifestation of, organism function, affects, clinical attribute)**

Exemplar prompt for generating GIVE$_a$:

You are a helpful assistant that answers a given question about medical knowledge with yes, no or maybe, based on the retrieved knowledge triplets (entity, relation, entity) from your own knowledge. The return must be one of yes, no or maybe.
[k-shot EXAMPLES]
Q: Traumatic aortic injury: does the anatomy of the aortic arch influence aortic trauma severity?
**[AFFIRMATIVE KNOWLEDGE TRIPLETS]**
eg: ('anatomical abnormality', 'affects', 'organism function'), ('injury or poisoning', 'affects', 'organism function'), ('anatomy', 'part of', 'aortic arch'), ('injury or poisoning', 'affects', 'organ or tissue function'), ('aortic arch', 'location of', 'injury or poisoning')...
**Output: maybe (GIVE$_a$)**
Logic Chain: I reached the answer 'maybe' by considering the relationship between the anatomy of the aortic arch and the severity of aortic trauma. The knowledge triplets suggest that the anatomy of the aortic arch may influence the severity of aortic trauma, as anatomical structure correlates with clinical attributes and impacts clinical attributes. Additionally, the severity of aortic trauma may correlate with clinical attributes, which can be affected by traumatic aortic injury. However, the relationship between the anatomy of the aortic arch and the severity of aortic trauma is not definitively stated in the knowledge triplets, hence the answer 'maybe'.

Exemplar prompt for generating GIVE$_{a+c}$:

You are a helpful assistant that answers a given question about medical knowledge with yes, no or maybe, based on the retrieved knowledge triplets (entity, relation, entity) from your own knowledge.
[k-shot EXAMPLES]
Q: Traumatic aortic injury: does the anatomy of the aortic arch influence aortic trauma severity?
**[AFFIRMATIVE KNOWLEDGE TRIPLETS]**
eg: ('anatomical abnormality', 'affects', 'organism function'), ('injury or poisoning', 'affects', 'organism function'), ('anatomy', 'part of', 'aortic arch'), ('injury or poisoning', 'affects', 'organ or tissue function'), ('aortic arch', 'location of', 'injury or poisoning')...
A: maybe (GIVE$_a$)
**Additional knowledge triplets: [COUNTER-FACTUAL KNOWLEDGE TRIPLETS]**
eg: ('organism', 'not result of', 'aortic trauma severity'), ('injury or poisoning', 'not complicates', 'anatomical structure'), ('aortic arch', 'not influence', 'injury or poisoning')...
**Output: yes (GIVE$_{a+c}$)**
Logic Chain: By utilizing the additional knowledge triplets provided, it can be inferred that the severity of aortic trauma is not a result of an experimental model of disease, steroid, traumatic aortic injury, or injury or poisoning. This suggests that the severity of aortic trauma is not influenced by these factors. Therefore, the anatomy of the aortic arch may indeed influence the severity of aortic trauma in cases of traumatic aortic injury.

Exemplar prompt for generating GIVE$_{a+c+e}$:

You are a helpful assistant that answers a given question about medical knowledge with yes, no or maybe, based on the retrieved knowledge triplets (entity, relation, entity) from your own knowledge, and the knowledge triplets from an expert knowledge base. The return must be one of yes, no or maybe.

[k-shot EXAMPLES]

Q: Traumatic aortic injury: does the anatomy of the aortic arch influence aortic trauma severity?

**[AFFIRMATIVE KNOWLEDGE TRIPLETS]**

eg: ('anatomical abnormality', 'affects', 'organism function'), ('injury or poisoning', 'affects', 'organism function'), ('anatomy', 'part of', 'aortic arch'), ('injury or poisoning', 'affects', 'organ or tissue function'), ('aortic arch', 'location of', 'injury or poisoning')...

A: maybe (GIVE$_a$)

**Additional knowledge triplets: [COUNTER-FACTUAL TRIPLETS]**

eg: ('organism', 'not result of', 'aortic trauma severity'), ('injury or poisoning', 'not complicates', 'anatomical structure'), ('aortic arch', 'not influence', 'injury or poisoning')...

A: yes (GIVE$_{a+c}$)

**Additional knowledge triplets retrieved from expert knowledge base: [EXPERT KG KNOWLEDGE TRIPLETS]**

eg: ('injury or poisoning', 'result of', 'anatomical abnormality'), ('steroid', 'causes', 'injury or poisoning'), ('injury or poisoning', 'complicates', 'anatomical abnormality'), ('anatomical abnormality', 'result of', 'injury or poisoning')...

**Output: yes (GIVE$_{a+c+e}$)**

Logic Chain: I utilized the external knowledge from the expert knowledge base to modify my previous answers by considering the additional knowledge triplets that were retrieved. Aortic arch is part of anatomical structure, which is the location of injury or poisoning. Anatomical structure also affects cell function, which further affects injury or positioning, suggesting the anatomy of the aortic arch could influence the severity of aortic trauma. Therefore, the correct answer to this question should be 'yes'.

