# OpenReview forum: "GIVE: Structured Reasoning of Large Language Models with Knowledge Graph Inspired Veracity Extrapolation"
_ICML.cc/2025/Conference — ICML 2025 poster_

### Official Review · Reviewer_aT9E · 2025-03-13

**Overall Recommendation:** 2

**Summary:**

The authors propose GIVE, an innovative framework designed to enhance the performance of large language models in scientific reasoning tasks. The framework consists of three main stages: expert data observation, divergent thinking, and information synthesizing. The large language model constructs a structured reasoning path by combining its internal parameterized knowledge with external non-parametric knowledge through a knowledge graph-inspired method called veracity extrapolation. This approach reduces hallucinations by utilizing counterfactual knowledge. It significantly improves the model’s accuracy and interpretability in biomedical, commonsense, and open-domain reasoning tasks, achieving an efficient balance in integrating large language models with knowledge sources that are either limited or noisy. As a result, it delivers superior reasoning outcomes.

**Claims And Evidence:**

Firstly, the authors propose that the introduction of expert knowledge improves reasoning accuracy, as demonstrated in the experimental analysis in Section 4.7.3. However, there seems to be an issue with the experimental data. Figure 4 shows that as the expert knowledge ratio increases, the average accuracy gradually improves, eventually reaching 100%. On one hand, the authors do not specify which backbone models are used for the data shown in Figure 4, and the full experimental details are not provided. On the other hand, according to the results, an injection of around 20% expert knowledge already leads to an average accuracy of 100%, which seems contrary to common sense. It would be helpful for the authors to provide further clarification on this part.

Secondly, the authors propose that GIVE can handle limited external knowledge bases, as demonstrated in Sections 4.4 and 4.5. However, in Table 4 of Section 4.5, there is no significant difference between GIVE, RAG, and ToG methods when providing 10%, 50%, and 100% scale knowledge graphs. It is possible that for commonsense reasoning tasks, the knowledge graph ConceptNet does not play a major role, and this argument is hard to substantiate. It is suggested that the authors provide experimental data for the 0% scenario for further analysis.

**Essential References Not Discussed:**

The key contribution is overcoming the limitations of relying solely on internal or external knowledge and fully utilizing both external knowledge bases and the model’s existing knowledge. This has been preliminarily explored in the EMNLP 2024 paper "Chain-of-Note: Enhancing Robustness in Retrieval-Augmented Language Models," which uses the model's existing knowledge to assess the relevance and validity of external knowledge, thereby improving answer accuracy. It is suggested that the authors include a discussion on models of this type of RALM.

[1] Yu, Wenhao, et al. "Chain-of-note: Enhancing robustness in retrieval-augmented language models." arXiv preprint arXiv:2311.09210 (2023).

**Experimental Designs Or Analyses:**

Firstly, Figure 4 shows that as the expert knowledge ratio increases, the average accuracy gradually improves, eventually reaching 100%. On one hand, the authors do not specify which backbone models are used for the data presented in Figure 4, and the full experimental details are not provided. On the other hand, according to the results, an injection of around 20% expert knowledge already leads to an average accuracy of 100%, which seems contrary to common sense. It would be helpful for the authors to provide further clarification on this part.

Secondly, in Table 4, regardless of whether it is GIVE, RAG, or ToG methods, there is no significant difference when providing 10%, 50%, and 100% scale knowledge graphs. It is possible that for commonsense reasoning tasks, the knowledge graph ConceptNet does not play a major role, and this argument is hard to substantiate. It is suggested that the authors provide experimental data for the 0% scenario for further analysis.

**Methods And Evaluation Criteria:**

From the main experimental results in Table 3, it can be observed that in the biomedical dataset selected by the authors, overall, the performance of GIVE_a+c is better than that of GIVE_a+c+e. This indicates that the introduction of expert knowledge may not only be ineffective but could even lead to a decrease in performance. Meanwhile, results on CommonsenseQA shown in Table 4, the difference between GIVE_a+c and GIVE_a+c+e is within the range of 0.1 to 0.5. In the proposed framework, the introduction of expert knowledge is an important part, and it would be helpful if the authors could provide further clarification on this matter.

**Other Comments Or Suggestions:**

If the example provided in Figure 2 could demonstrate connecting all four sets of entities, it might be more helpful for readers to understand. Currently, it shows cross-group connections with two sets per group.

**Other Strengths And Weaknesses:**

Please refer to Questions For Authors

**Questions For Authors:**

1.Discussion of RALM Models in the Context of Your Framework

The key contribution of your paper involves overcoming the limitations of relying solely on internal or external knowledge, fully utilizing both external knowledge bases and the model’s existing knowledge. This concept has been explored in the EMNLP 2024 paper "Chain-of-Note: Enhancing Robustness in Retrieval-Augmented Language Models." Could you include a discussion on models of this type of RALM and how they relate to your framework?

2.Role of Expert Knowledge in Cross-Group Connections

The paper suggests that "The expert's cross-group connections serve as evidence, guiding the LLM to extrapolate the veracity of potential relationships among similar concepts." However, there seems to be insufficient explanation of the direct role of expert knowledge in establishing cross-group connections. Could you clarify how expert knowledge directly contributes to this process?

3.Performance of GIVE_a+c vs. GIVE_a+c+e in Biomedical Dataset

In Table 3, the performance of GIVE_a+c is better than that of GIVE_a+c+e in the biomedical dataset. Can you provide an explanation as to why the introduction of expert knowledge may lead to a decrease in performance in this case?

4.Expert Knowledge Injection and Accuracy

The results in Figure 4 show that injecting around 20% expert knowledge leads to an average accuracy of 100%. This seems counterintuitive and contrary to common sense. Could you provide further clarification on this, and explain why such a small amount of expert knowledge leads to such high accuracy?

5.Clarification on Backbone Models and Experimental Details

Could you specify which backbone models are used for the data presented in Figure 4 and provide full experimental details? The lack of such information raises concerns about the reliability and reproducibility of the results.

6.Discrepancy Between GIVE, RAG, and ToG in Table 4

In Table 4, there seems to be no significant difference between GIVE, RAG, and ToG methods when providing 10%, 50%, and 100% scale knowledge graphs. Could you explain why the knowledge graph ConceptNet does not show a significant effect, especially for commonsense reasoning tasks? It would also be helpful if you could provide experimental data for the 0% scenario for further analysis.

**Relation To Broader Scientific Literature:**

N/A

**Theoretical Claims:**

The paper doesn't present complex theoretical proofs but focuses on algorithmic contributions.

---

> ### Author Rebuttal · Authors · 2025-03-31
>
> We understand the reviewer's concerns about the role of expert knowledge in the reasoning process, as well as in the answer-generation process, and we appreciate your suggestion on our literature discussion and experiments. We hope the following clarifications addressed all the questions and we will make sure to include these discussions in our revised manuscript.
>
> > For Claims And Evidence paragraph 1:
>
> We apologize for the confusion. To clarify, GIVE enriches the expert knowledge in the KG by extrapolating expert knowledge towards relevant concepts akin to the queried ones, as detailed in Section 3.4.3. Direct injection of expert knowledge into the question-answering is $\textbf{not}$ assumed, due to the uncertain quality of the accessible knowledge base. Figure 4 shows that the performance of GIVE improves with an increased ratio of expert knowledge to total knowledge (expert plus extrapolated). Higher expert knowledge ratio ensures that the neighborhood related to the query is well-connected in the KG, offering substantial evidence for veracity extrapolation. Lower ratio indicates reliance on internal knowledge,  as discussed in Section 3.3 and Open Relations in Section 3.4.3. GIVE performs well with ample expert evidence guiding reasoning, quantitatively when expert knowledge ratio reaches about 20\%, $\textbf{not}$ when injecting solely 20\% of this knowledge. All ablation studies in Section 4.7 utilize GPT3.5-turbo, with other details consistent with Section 4.2.
>
> > For Claims and Evidence Paragraph 2:
>
> Commonsense reasoning experiments aim to test GIVE's generalizability in three ways: (1) on noisy, large-scale KG, (2) varying sparsity levels, and (3) tasks where pre-trained knowledge is extensive. This setting presents challenges, as the basic LLM already achieves about 70\% accuracy; pre-training on commonsense knowledge is inherently easier than on scientific information. In such cases, misinformation can lead to hallucination. GIVE improves performance by up to 3.4\% and 4.9\% over RAG and ToG, and consistently outperforms base LLM, indicating its reasoning process avoids hallucinations. ConceptNet is indeed important, the challenge is how to wisely use its information to further enhance the rich pre-training knowledge without causing hallucination, especially with its sparse versions.
>
> Including a 0\% scenario, as the reviewer suggested, offers more insight: GIVE_a and GIVE_a+c achieve 69.84\% and 69.36\%, respectively, due to the exclusion of expert information. Without expert information, the knowledge provided by GIVE is derived solely from the LLM's internal knowledge (Sections 3.3 and Open Relations in 3.4.3). This aligns with our observation in Figure 4, where the expert knowledge ratio is 0. The better performance in this case is attributable to richer pre-training in commonsense compared to biomedical knowledge.
>
> > For Methods and Evaluation Criteria:
>
> We understand the reviewer's concern about the role of expert knowledge. GIVE is designed for situations where a high-quality KG is inaccessible in domain-specific tasks. In such scenarios, GIVE does $\textbf{not}$ intend to utilize expert knowledge directly for question answering but rather as an "inspiration," illustrated by the transition from solid to dashed lines in Figure 2. The KG links some entities unrelated to the query; GIVE encourages the model to assess whether similar connections exist among other related entity pairs. In Tables 3 and 4, GIVE_a+c+e generally demonstrates improved or equivalent performance. The minor margin does not imply that expert information is trivial, but rather reflects that they are not directly solving the query. Expert knowledge (solid lines in Figure 2) is pivotal for the success of GIVE_a and GIVE_a+c, as it contributes to knowledge extrapolated from expert connections among relevant entities.
>
> > For Essential References Not Discussed:
>
> We will include discussion of CoN in our revised manuscript. GIVE and CoN differ fundamentally in several aspects: (1) Motivation: GIVE is a reasoning framework that formulates a faithful thinking process by populating the expert KG triplets towards the query, whereas CoN is a robust retrieval system excluding similar yet irrelevant documents. (2) Use of internal knowledge: GIVE employs LLM's internal knowledge for "veracity extrapolation," depicted by the transition from solid to dashed lines in Figure 2, whereas CoN uses it to create document summaries (notes) for accurate relevance analysis. (4) Use of external knowledge: All expert knowledge in GIVE is integral to its reasoning process, as shown by the solid lines in Figure 2. Meanwhile, CoN filters out irrelevant documents for question-answering. (5) GIVE is designed to reason $\textbf{beyond}$ the accessible KG, whereas CoN identifies and focuses on documents containing the essential context.
>
> > For Other Comments Or Suggestions:
>
> We will include all connections in the Figure for our revised manuscript.

---

### Official Review · Reviewer_6XQ9 · 2025-03-14

**Overall Recommendation:** 3

**Summary:**

The paper introduces Graph Inspired Veracity Extrapolation, a reasoning framework that enhances LLMs by integrating parametric and non-parametric memories for more accurate reasoning with minimal external input. GIVE operates through three key steps to select relevant expert data, engage in query-specific divergent thinking, and synthesize information for final outputs. Extensive experiments show that GIVE improves LLM performance across different model sizes, allowing smaller models to outperform larger ones in scientific tasks. GIVE supports reasoning with both restricted and noisy knowledge sources and underscores the value of combining internal and external knowledge to enhance LLM reasoning capabilities for complex scientific tasks.

**Claims And Evidence:**

The claims made in the paper are supported by experimental evidence. The authors demonstrate through extensive experiments that GIVE improves LLM performance across various sizes and domains.

**Essential References Not Discussed:**

1. HOLMES:Hyper-Relational Knowledge Graphs for Multi-hop Question Answering using LLMs
2. KAG:Boosting LLMs in Professional Domains via Knowledge Augmented Generation

**Experimental Designs Or Analyses:**

The proposed approach is only compared with naïve RAG and GraphRAG. However, many graph-based RAG methods have recently emerged, with their code publicly available. It would be beneficial to include comparisons with these methods, such as KAG and HOLMES, to provide a more comprehensive evaluation.

**Methods And Evaluation Criteria:**

The proposed methods make sense for improving LLM reasoning in scenarios where internal knowledge is insufficient and external knowledge is limited or noisy. The evaluation criteria, including accuracy on various reasoning tasks, are appropriate for assessing the effectiveness of the proposed framework. The use of both scientific and open-domain datasets provides a comprehensive evaluation of the method's applicability.

**Other Comments Or Suggestions:**

1. The proposed approach is compared only with naïve RAG and GraphRAG. However, several recent graph-based RAG methods, with publicly available code, have been introduced. Including comparisons with methods like KAG and HOLMES would provide a more comprehensive evaluation of the approach.
2. It seems that the BioASQ dataset is highly sensitive to the retrieved knowledge. A more detailed discussion and analysis of this sensitivity would be valuable.
3. While time consumption is provided in the supplementary material, a more in-depth analysis of token consumption would be helpful.

**Other Strengths And Weaknesses:**

Weaknesses
The token usage and time consumption are significantly higher than other methods, yet the paper provides limited discussion on token efficiency, potential optimizations, and the impact on scalability and real-world applications.

**Questions For Authors:**

please refer the suggestions and weakness.

**Relation To Broader Scientific Literature:**

The key contribution of this paper is: This work proposes a structured reasoning framework that integrates internal and external knowledge, along with a veracity extrapolation method that enriches limited information by establishing provisional connections between query concepts and incorporating counterfactual reasoning to mitigate hallucinations.

**Theoretical Claims:**

No theoretical claims are provided.

---

> ### Author Rebuttal · Authors · 2025-04-01
>
> > In-depth analysis of token consumption
>
> We understand the reviewer's concern about the token efficiency and provide a comparison of token consumption between GIVE and ToG on 100 random questions from each dataset. For each question, we calculate the total number of input/output token in the whole problem-solving process. We use tiktoken with GPT3.5-turbo for this comparison. For GIVE we use n=1, and for ToG, we use D=5, the setting is the same as Table 8 in the Appendix.
>
> |Avg no.input tokens        | PubmedQA | BioASQ | ProcessBank | CSQA/10\% ConceptNet | CSQA/50\% ConceptNet | CSQA/100\% ConceptNet |
> |--------|----------|----------|----------|----------|----------|----------|
> | GIVE  | 14518.5     | 7970.3     | 19460.5    | 5321.1     | 7203.0    | 7398.7     |
> | ToG  | 12701.1    | 7010.0     | 11995.6     | 4934.8    | 6704.1    | 6679.7    |
>
> |Avg no.output tokens        | PubmedQA | BioASQ | ProcessBank | CSQA/10\% ConceptNet | CSQA/50\% ConceptNet | CSQA/100\% ConceptNet |
> |--------|----------|----------|----------|----------|----------|----------|
> | GIVE  | 183.1     | 80.0     | 232.5     | 34.9     | 45.2     | 46.3     |
> | ToG  | 104.2    | 60.2     | 100.5     | 23.1     | 34.8     | 36.6    |
>
> The observation supports the conclusion of Table 8, where GIVE\_n=1 effectively balances efficiency and accuracy in challenging scientific reasoning tasks. Specifically, in 5 out of 6 datasets, GIVE consumes around 10\% more input tokens than ToG, generates just 80 and 20 more tokens for PubmedQA and BioASQ, respectively, while reaching a 3-fold and 5-fold increase in accuracy. The variance in token usage across datasets is due to the differing numbers of entity groups and numbers of candidate relations, leading to varying candidate connections, as depicted in Figure 6 of the appendix. GIVE also demonstrates strong generalizability in deployment on KGs with varying sizes and sparsities.
>
> > Discussion of KAG and HOLMES
>
> Thank you for pointing out the related works; we will add a discussion of KAG and HOLMES in our revised manuscript. Although we acknowledge that KAG and HOLMES leverage KG to enhance LLM output, we respectfully contend that they are solving fundamentally different problems with GIVE.
>
> KAG and HOLMES are advanced $\textbf{retrieval}$ systems that retrieve and integrate accurate expert knowledge in question-answering tasks. In contrast, GIVE, as a $\textbf{reasoning}$ framework, uses irrelevant information pieces to promote divergent thinking and extrapolates knowledge to generate answers. Thus, KAG and HOLMES concentrate on extracting quality KG from corpus or documents to aid precise information matching via a refined SemanticGraph or Graph Schema. They use KG for more accurate information matching. Here, the LLM iteratively synthesizes the responses as an answer generator. GIVE, however, guides LLMs in problem-solving like a human expert, foregoing preprocessing irrelevant knowledge by using the structured nature of KGs to seamlessly connect expert information with a query. KG data (entities and their connections) support the construction of a reasoning chain. LLM solves queries by verifying the inferred links suggested by KG, as illustrated in Figure 2. Although we were committed to including them as baselines, as suggested by the reviewer, HOLMES currently lacks an open-source implementation, and the public KAG code lacks the kag-model, rendering them unusable in our experimental settings with query and KG as inputs. We believe that these methods will perform similarly to GraphRAG, emphasizing information retrieval and injecting over reasoning.
>
> > Discussion of sensitivity to retrieved knowledge
>
> We appreciate your insightful discussion. To clarify, the experiments in Figure 4 consistently show that GIVE effectively uses expert knowledge not directly related to the question to create an accurate reasoning process. This is evidenced by an improvement in performance from n = 0 to n = 1 and improved results with a higher ratio of expert knowledge. Furthermore, with the same expert knowledge ratio, BioASQ performs better as its questions are more easily connected to expert knowledge. The veracity extrapolation process links expert knowledge to the query by the extrapolated connections; a lower expert knowledge ratio means fewer such "bridges" for the model to leverage the unrelated KG information. We will incorporate this discussion into our revised manuscript.

---

### Official Review · Reviewer_mrWr · 2025-03-14

**Overall Recommendation:** 5

**Summary:**

This paper proposes a novel reasoning method based on LLMs and an external knowledge graph. The motivation of this paper is to address the reasoning quality under the situations that the knowledge graph is sparse. This is reasonable as CoT has no external knowledge, while RAG and ToG suffer from the scale of the external base knowledge. To address these challenges, the authors first transfer the queries into entities and relations formed in knowledge graphs, and then construct the entity groups by computing the cosine similarities in the embedding space. After this, the authors execute the inner-group to filter broader related concepts and inter-group connections to link both KG relations between groups and question related relations. The experiments show significant improvements in biomedical QA and robustness and scalability.

## update after rebuttal
I would like to maintain the current rating.

**Claims And Evidence:**

Yes, the claims that the authors made are intuitively correct. For example, the authors list the challenges of CoT, RAG, ToG and show why they are unworkable with a sparse knowledge base graph. The visualizations are clear to illustrate this issue with a typical example.

**Essential References Not Discussed:**

The literature is sufficient.

**Experimental Designs Or Analyses:**

The experiments are sufficient to answer five questions provided by the authors in Section 4.1.

**Methods And Evaluation Criteria:**

Yes, the method is reasonable. The benchmark datasets are sufficient to support the claims.

**Other Comments Or Suggestions:**

Typos:
- line 86, 'focuses on uses’ -> 'focuses on using’.
- line 192, ',The set’.

**Other Strengths And Weaknesses:**

Other Strengths:
- The authors provide a comprehensive method for LLM reasoning that significantly improves the LLM reasoning quality when the knowledge graph is large and the information is sparse. This setting is widely existing in the industry.
- The paper is well-written with clear visualizations.
- The authors have conducted lots of experiments. Particularly, the authors test the performance of GIVE on knowledge graphs with scales from 135 entities to 844K, which validates GIVE’s capability of retrieving on both small and large knowledge graphs.

Other Weaknesses:
Overall, I do not find severe problems in this paper, the following could be some `weaknesses’ that the authors or other reviewers might concern:
- Compared to RAG that can retrieve any text information, graph-based retrievers (also for GraphRAT, etc.) naturally require an external knowledge base graph that is of high quality and is typically not yet learned by the LLMs. This might not be a `weakness’ but the common limitation of graph-based retrievers.
- GIVE requires the preprocessing on the knowledge graph, the extraction of query information, and retrieval, which take some time. Some of the processes are before inference (like preprocessing), while others (extraction and retrieval) are during inference.
- The method introduces some additional hyperparameters like m and n, increasing the difficulty of adjusting hyperparameters.

**Questions For Authors:**

- Since I did not find the code link, I just want to ask the authors whether they would provide the corresponding repository to the public?

**Relation To Broader Scientific Literature:**

This work is in terms of the LLM reasoning field. Related works include Chain-of-Thoughts (CoT), Retrieval-Augmented-Generation (RAG), GraphRAG, etc.

**Theoretical Claims:**

No theoretical claim is provided.

---

> ### Author Rebuttal · Authors · 2025-03-31
>
> > Compared to RAG that can retrieve any text information, graph-based retrievers (also for GraphRAT, etc.) naturally require an external knowledge base graph that is of high quality and is typically not yet learned by the LLMs. This might not be a `weakness’ but the common limitation of graph-based retrievers.
>
> We appreciate the recognition of our contributions and agree with the common limitation of graph-based retrievers as mentioned by the reviewer. GIVE plays a pivotal role in addressing the challenge where high-quality Knowledge Graphs (KG) are not readily available. Knowledge Graphs possess certain distinct advantages over textual corpora. The primary goal of GIVE is to bridge the gap between the limited parametric knowledge typically available and the accurate reasoning required for scientific queries by effectively populating limited expert information towards the query. Our choice of KGs as a data source is due to their structured relational knowledge, which naturally provides a reasoning pathway. This choice simplifies the task to effectively creating connections between the expert KG knowledge with the ultimate question-answering task. In such scenarios, the process of constructing a relevant entity set and retrieving the connections between different groups proves efficient due to the inherent structure of graph data. Recent studies [1] have focused on converting text into KGs, and we are optimistic that textual corpora and KGs will become mutually interchangeable with advancements in this field of research.
>
> [1] KGGEN: EXTRACTING KNOWLEDGE GRAPHS FROM PLAIN TEXT WITH LANGUAGE MODELS.
>
>
> > GIVE requires the preprocessing on the knowledge graph, the extraction of query information, and retrieval, which take some time. Some of the processes are before inference (like preprocessing), while others (extraction and retrieval) are during inference.
>
> We appreciate the constructive feedback from the reviewer. We agree with the reviewer's insight that the reasoning process used by GIVE demands additional inference time, particularly in the construction of entity groups and the extrapolation of veracity. To substantiate our claims, we present comprehensive experiments in Table 8 in the appendix. These experiments demonstrate that when the number of entities per group (n) is set to 1, GIVE surpasses other competing baselines in performance, while maintaining similar or even lesser execution times. Notably, this trend is more significant when applying larger and denser Knowledge Graphs (KGs). Efficiency of GIVE could be further improved by incorporating batch pruning for the veracity extrapolation process. Moreover, as elaborated in Section 5, future research could involve automating the "veracity extrapolation" procedure, for instance, by developing a corresponding process reward that can be incorporated into Reinforcement Learning (RL) post-training frameworks. We are confident that the contributions of GIVE offer substantial benefits for future endeavors in this area of research.
>
> > The method introduces some additional hyperparameters like m and n, increasing the difficulty of adjusting hyperparameters.
>
> We appreciate your effort in highlighting this issue. It is important to elaborate further that GIVE's only hyperparameter is the number of entities per group, denoted as \( n \). In contrast, \( m \) represents the number of entity groups which is a characteristic inherent to the query, as explained in Sections 3.1 and 3.2. For each queried entity, an entity group is created using its analogous concepts in the Knowledge Graph (KG). The findings illustrated in Figure 4 and Table 8 substantiate that GIVE offers commendable performance even with \( n = 1 \). This facilitates GIVE in achieving an optimal balance between efficiency and accuracy without the necessity of hyperparameter tuning. Moreover, experiments detailed in Appendix C.1 reveal that the average value of \( m \) is typically 3 or 4 across all datasets.
>
> > Since I did not find the code link, I just want to ask the authors whether they would provide the corresponding repository to the public?
>
> Thank you for raising this issue. We hold open-source research in high regard and are committed to including a link to the repository containing all the relevant codes and datasets in the revised manuscript.

---

### Official Review · Reviewer_JLdU · 2025-03-25

**Overall Recommendation:** 4

**Summary:**

This paper introduces GIVE to facilitate the reasoning ability of LLMs in specific domains. GIVE extracts the relevant information of a knowledge graph and bridges it with the query by using LLM’s internal knowledge to justify the veracity of the extrapolated knowledge. GIVE also incorporates counterfactual knowledge and progressive answer generation to alleviate hallucination caused by the additional knowledge. The authors conducted extensive experiments using both domain-specific and open-domain benchmarks, utilizing KGs of various sizes and sparcities. The empirical results proved the effectiveness and generalizability of GIVE.

**Claims And Evidence:**

Yes

**Essential References Not Discussed:**

no

**Experimental Designs Or Analyses:**

Yes

**Methods And Evaluation Criteria:**

Yes

**Other Comments Or Suggestions:**

no

**Other Strengths And Weaknesses:**

Strength:

1. The idea is novel. GIVE innovates in directing LLM to further populate the retrieved information to formulate a multi-step reasoning chain, differing from the reasoning methods on self-knowledge or the retrieval methods for gold context.

2.  Experiments are comprehensive. In addition to various benchmarks, the author includes ablation studies to study the sensitivity of GIVE to parameters, and “expert knowledge ratio”, These findings further support the intuition of GIVE.

3. Analysis is sufficient. The Supp further validates the claim on the efficiency of GIVE with detailed context length and accuracy of the proposed method with different parameters, providing insights to the community.

Weakness:

1. From Figure 4 we see that the expert knowledge set plays an important role for GIVE. However, from Table 8, GIVE achieves
better performance than ToG and RAG when using n=1 and only the affirmative knowledge set, why can GIVE achieve good performance without using the expert knowledge set?

2. In Table 3, why does CoT harm the performance of GPT4, as CoT does not introduce new knowledge which causes hallucination, as clarified by the authors.

3. In Figure 3, there is a huge performance gap between GIVE_a and GIVE_a+c, GIVE_a+c+e. Can the authors clarify the cause of this performance gap

**Questions For Authors:**

no

**Relation To Broader Scientific Literature:**

Previous works on LLM with external knowledge integration heavily focuses on the task of KGQA, where the gold answer or reasoning path is contained in the given KG, which is diMerent from the research problem defined in this paper. GIVE specializes in using limited information to prompt LLM reasoning in the specific domains, the authors include suMicient discussion of the previous research in the introduction, the motivation to propose GIVE for the hard tasks where retrieving from high-quality knowledge sources is clear.

**Theoretical Claims:**

Yes

---

> ### Author Rebuttal · Authors · 2025-03-31
>
> > From Figure 4 we see that the expert knowledge set plays an important role for GIVE. However, from Table 8, GIVE achieves better performance than ToG and RAG when using n=1 and only the affirmative knowledge set, why can GIVE achieve good performance without using the expert knowledge set?
>
> We are grateful for the insightful discussion brought up by the reviewer. It is essential to further elucidate that GIVE employs the expert knowledge extracted from the Knowledge Graph (KG) with the purpose of enabling faithful reasoning through knowledge extrapolation, rather than being directly utilized for answer generation. As demonstrated in the experiments reflected in Figure 4, the findings indicate that the efficacy of GIVE's "veracity extrapolation" is intricately linked to the ratio of expert knowledge. This ratio evaluates the degree to which the KG is pertinent to the questions, encompassing both the entities and their interrelations. The reason for this relationship is that expert knowledge constitutes the KG's connections among the pertinent entities, as detailed in Section 3.2 and depicted by the solid lines in Figure 2 of our initial manuscript. Although this expert knowledge might not directly facilitate the answering of questions, as it is not immediately connected to the queries, it serves as a valuable source of "inspiration" for the model to execute divergent reasoning. This is a primary rationale behind proposing a framework that integrates retrieval with reasoning when dealing with external knowledge that is sparse and noisy.
>
>
>
> > In Table 3, why does CoT harm the performance of GPT4, as CoT does not introduce new knowledge which causes hallucination, as clarified by the authors.
>
> We appreciate your question. In Section 4.4, we critically evaluate various approaches by assessing their performance on scientific benchmarks that demand intensive reasoning. Large language models (LLMs) lack sufficient parametric knowledge to effectively address these complex queries. This is primarily because it is impractical to extensively train these models on highly specialized domains. Our observations, derived from comparing the outcomes of GPT-3.5-turbo and GPT-4 on datasets such as PubMedQA and ProcessBank, illustrate that simply scaling up pre-training does not necessarily enhance accuracy. In such instances, the Chain-of-Thought (CoT) approach seeks to promote a logical step-by-step reasoning process that relies exclusively on the model's internal knowledge, thus results in fluctuating performance levels due to its inherent randomness.
>
> > In Figure 3, there is a huge performance gap between GIVE_a and GIVE_a+c, GIVE_a+c+e. Can the authors clarify the cause of this performance gap
>
> Thank you for raising this question. Figure 3 illustrates our experiments with TruthfulQA, an open-domain text generation task. This dataset is included to demonstrate that GIVE is effective across both domain-specific and open-domain tasks.  GIVE_a achieves superior results primarily because inquiries are directed at information accuracy. Consider the example query for TruthfulQA "What is the original meaning of SOS?" In this scenario, incorporating an affirmative knowledge set naturally leads to a more coherent context, whereas integrating counter-factual information tends to result in hallucinations. Additionally, as we employed the 10\% version of ConceptNet in this experiment, the retrieved expert knowledge is not directly associated with the query. Instead, it is utilized to motivate the model towards "veracity extrapolation," which significantly contributes to the excellent performance of GIVE_a. This is elaborated upon in Section 4.6 (lines 332-338), and the solid to dashed line process in Figure 2 of our original manuscript.

---

> > ### Comment · Reviewer_JLdU · 2025-04-03
> >
> > Thanks for the rebuttal from the authors, which well resolves my concerns. I therefore increase my score.

---

> > > ### Author Response · Authors · 2025-04-03
> > >
> > > Thank you for appreciating our work and your continuous engagement in the discussion. Your input has been important for us to improve the paper.

---

### Decision · Program_Chairs · 2025-05-01

**Decision:**

Accept (poster)

**Comment:**

The GIVE framework presented in this paper is designed to improve the reasoning capabilities of large language models (LLMs) by combining their internal knowledge with external knowledge graphs. This approach facilitates more accurate and efficient reasoning with minimal external input. The framework operates in three key stages: observation of expert data, divergent thinking, and synthesis of information to produce the final output. Extensive experiments have demonstrated the benefits of GIVE, including its ability to boost the performance of LLMs across various sizes, enable smaller models to outperform larger ones in scientific tasks, and maintain robustness and scalability in both scientific and open-domain assessments. GIVE is also highlighted for its ability to reason with both restricted and noisy knowledge sources, and its fully interpretable reasoning process.

While there are many discussions in the rebuttal period, the main opposing opinions including comparison with CoN, KAG and HOLMES, the analysis of token consumption, the role of expert knowledge in the reasoning process and the answer-generation process, etc.  The author addressed these questions with clarity and explanations.

The GIVE framework as presented in the paper has the potential impact to the fields by improving the reasoning capabilities of AI systems, making them more accurate, interpretable, and efficient, particularly in specialized domains where such capabilities are in high demand.